# Interpreting Kolmogorov-Arnold Networks in Neuroimaging: A Path-Based Attribution Framework

**Suhrud Murthy**[1]     **R. Venkatesh Babu**[2]     **Neelam Sinha**[1] *
[1] *Centre for Brain Research, IISc Bangalore*
[2] *Vision and AI Lab, Department of Computational and Data Sciences, IISc Bangalore*

**Reviewed on OpenReview:** *https: // openreview. net/ forum? id= cPtKpNdYc2*

## Abstract

Explainability aspects of most classification models are learnt through instance-specific analysis. However, in understanding diseases, it is important to consider population-wide analysis in order to identify affected regions that are consistently seen across cohorts of the diseased population. In this study, we report the utility of Kolmogorov-Arnold Networks (KANs) in understanding population-wide characteristics seen in subjects affected by Alzheimer's disease (AD). KANs offer enhanced interpretability through learnable activation functions on network edges. Thus, the learned functions reflect the characteristics of the entire span of training data. In a KAN network trained for classification, attributions through the network can be traced to understand how specific inputs influence the output label. In this study, we propose a path-based attribution framework that generates global importance maps by tracing exhaustive information flow through all potential paths. Our method initially scores the functions on the edges of a trained KAN using an appropriate scoring function. Subsequently, these scores are propagated through the network to compute path-attributions. This approach scales linearly with network depth, and is only dependent on model training and does not need further analysis on training data post-hoc. Evaluation on three public AD neuroimaging datasets (OASIS, ADNI, Mendeley, totally comprising 7428 acquisitions), were carried out on 2D brain slices as well as 3D brain volumes. The corresponding KAN test accuracies are 93.24%, 81.85%, and 91.25% on OASIS, ADNI, and Mendeley datasets, respectively. Alongside, competitive or improved performance via metrics such as Insertion AUC, Deletion AUC and Sufficiency, is also demonstrated. The generated attribution maps identify clinically meaningful regions including the body and genu of corpus callosum, corona radiata, bilateral caudate nuclei, medial prefrontal cortex and temporal lobe structures, aligned with established AD pathology literature. By providing voxel-level global attributions as network-intrinsic properties, our framework addresses a critical gap in AI interpretability and supports exploratory clinical analysis and model auditing of AI-assisted AD diagnosis systems.

## 1 Introduction

The intersection of medical imaging and computational neuroscience has been driven forward by the advent of artificial intelligence and deep learning. Imaging modalities such as magnetic resonance imaging (MRI), such as structural MRIs (sMRI) and functional MRIs (fMRI), among several others, provide spatial and temporal data at a very high resolution. This allows researchers to investigate the anatomy of the brain at fine granularity, assisting in early diagnosis and understanding of disorders Lundervold & Lundervold (2019). Deep learning models achieve high accuracies on brain imaging tasks, yet their opacity limits clinical adoption and regulatory approval in neuroscience Kelly et al. (2019), with model decisions often being inscrutable.

---

*Corresponding Authors: Neelam Sinha <`neelam@cbr-iisc.ac.in`>, R. Venkatesh Babu <`venky@iisc.ac.in`>.

Interpretability of a network is crucial in medical analysis. Unlike consumer domains, medical decisions require transparent reasoning, and clinicians must be able to trace model decisions to anatomically or functionally meaningful biomarkers, which can also aid in distinguishing between medically backed model decisions and erroneous correlations that can occur due to noisy or unclean data Wang et al. (2024) Salahuddin et al. (2022). Several attribution methods have been put forth to understand model decisions and generate input attributions. Localized attribution methods such as gradient saliency maps, Layer-wise Relevance Propagation, Integrated Gradients, and occlusion-based methods, among others, have contributed insights to brain imaging by highlighting which voxels or regions contribute to model predictions. However, these approaches often produce localized, sample-specific explanations, making it difficult to identify which features or regions are consistently important across training data. While such local attributions can indicate potential importance for individual cases such as tumour detection, where tumours are unique to the subject and it is difficult to generalize the location a tumour across subjects, they may provide limited clarity into patterns seen across brain populations for neurodegenerative diseases such as Alzheimer's disease (AD) and dementia, among others, which involve diffuse, population-level structural changes that are better captured through global attribution methods.

Additionally, arriving at potentially thousands of local attributions for individual data items can be difficult to interpret. Global feature attributions, which show how features influence the decision of a model across the entire training distribution, instead provide a consolidated view of feature importance that highlights systematic patterns learned by the model during training rather than instance-specific effects. This distinction is particularly important in medical imaging, for problems such as Alzheimer's detection, where clinicians and researchers need to understand which anatomical regions the model has learned to consistently rely upon across patient populations Munroe et al. (2024).

Global attributions can provide several advantages over localized attribution in medical imaging contexts. It enables the identification of biomarkers that are present across the patient distribution in training data. It also aids model debugging by revealing biases or spurious correlations by highlighting the point of focus of the model, enabling users to identify, with known ground-truth markers, if the model is identifying appropriate features. By accurately highlighting clinically meaningful features learnt by the model and demonstrating that models rely on these features alone for their decision, regulatory approval and clinical trust can be potentially gained. It provides complementary insights to physician diagnosis Hill (2024). It also bridges the communication gap between the AI and medical world by providing population-level insights that align with medical knowledge. By attributing global features not as an aggregated property of the inputs, but as an intrinsic property of the trained network itself, we can generate attribution maps that highlight medically relevant regions of the brain.

Our work utilizes the inherent functional transparency of the recently proposed Kolmogorov-Arnold Networks (KANs). KANs, as originally proposed in Liu et al. (2025), are designed to replace every scalar weight in a multi-layer perceptron (MLP) with a learnable univariate spline function on each edge, as opposed to a node. They leverage the Kolmogorov-Arnold Representation Theorem, which states that "any continuous high-dimensional function can be decomposed into a finite sum of univariate continuous functions", which has made it possible to express the transformation from an input to the output decision as an explicit graph of simple, human interpretable univariate functions on edges that are simply added at each node. Due to the functions on each edge being explicit and univariate, it is possible to analyze the nature of each function and thereby quantify the influence of the function locally. Specifically, a learned function on an edge consists of a base component and a spline component, expressed as $\phi(x) = w_b b(x) + w_s spline(x)$, where $b(x)$ is typically the SiLU activation function and $spline(x)$ is a linear combination of basis splines. This structure makes it possible to identify the individual contributions of these functions, their interactions with the rest of the network, and their association with each individual input.

In neuroimaging, specifically in understanding the contribution of each individual voxel corresponding to a certain part of the brain, input-based attribution can prove to be particularly useful. By tracing the decision of the model back to a particular input/set of inputs, it can be made possible to identify and rank

the features influencing the model. This makes it possible to construct faithful[1], global attribution maps at input-level granularity by analyzing the flow of information through the network, which tell us about the parts of the brain the model has learnt to differentiate between.

In this work, we propose a path-based attribution approach for KANs to generate population-level importance maps as a property of the trained network, by tracing information flow along explicit computational paths, explicitly represented by a KAN's architecture, thereby scoring input importances based on the information flow. We also evaluate its utility and reliability for neuroimaging and clinical neuroscience applications. We demonstrate that this approach allows for faithful, global attribution maps that can provide clinically and anatomically useful information.

**Scope and focus.** The proposed framework is designed for analysis of fully trained feedforward networks in which nonlinear transformations are associated with edges, nodes aggregate incoming signals additively or multiplicatively, and directed paths exist from input features to output neurons. KANs constitute a prime example of architectures that satisfy these conditions, and hence we define and evaluate the framework as such. In this work, we focus on fully trained KANs, which we apply to neuroimaging tasks due to their high dimensionality and established interpretability challenges, and additionally evaluate the framework on a multiclass classification task using MNIST to assess behaviour in a simpler, well-controlled setting. We list our primary contributions as follows:

   i) A path-based attribution framework for KANs aggregating importance across computational paths.

   ii) A scalable implementation of path-attribution for data such as neuroimaging.

  iii) Voxel-level global-attribution maps applied through KANs for Alzheimer's detection.

  iv) A comprehensive clinical evaluation showing anatomically meaningful regions associated with AD, supported by ROI analyses.

## 2 Related Literature

Several attribution methods have been proposed to understand the functioning of neural networks as "black boxes". Gradient-based approaches, Simonyan et al. (2013) calculate output gradients with respect to inputs, and Sundararajan et al. (2017) integrate gradients along a baseline path to address gradient saturation. Propagation-based approaches such as Layerwise Relevance Propagation (LRP) Bach et al. (2015) have been applied to Alzheimer's detection, Böhle et al. (2019) using LRP on ADNI sMRI, producing maps concentrated in the temporal lobe and hippocampus. However, the authors note a high inter-patient variability in patterns, which is a limitation in generalizing patterns across the data population. Class Activation Mapping (CAM), particularly Eigen-CAM and Grad-CAM have also been adopted for model decision explanations, but fail to provide faithful explanations, often cover unnecessarily large or imprecise regions, exhibit low granularity in highlighting salient features, and suffer from instability and biases due to architectural factors such as Global Average Pooling and gradient explosion Bae et al. (2020). Game-theoretic approaches such as LIME, DeepLIFT Shrikumar et al. (2017) are also popular, with SHAP being implemented by Qiu et al. (2022) on the NACC dataset and OASIS cohort and found distinct patterns unavailable through averaging local explanations, which were negative hippocampal values for cognitively normal subjects, positive for AD patients. The problem with trying to arrive at a global explanation from these local attribution methods is it conflates intrinsic feature importance and feature distribution in data Covert et al. (2020). A feature may accumulate high average SHAP scores simply because it varies widely in training data, not because the model relies on it for decisions. This way, global maps can differ significantly from averaged local attributions Ibrahim et al. (2019), and this theoretical insight explains the empirical observation: across all local attribution methods (LRP, IG, Grad-CAM), importance scores fluctuate dramatically between patients with identical diagnoses, reflecting both model variance and data noise.

---

[1]We use the term "faithful" to denote how closely the produced attribution maps reflect the features the model has learned and utilizes to arrive at its decisions, with attribution quality quantified using perturbation-based metrics such as Insertion AUC, Deletion AUC, and Sufficiency.

To solve this problem, global attribution methods that highlight locations of important features seen throughout training data have been proposed. SAGE (Shapley Additive Global importancE) Covert et al. (2020) directly measures the contribution of features to the model's prediction by introducing features into arbitrary subsets and calculating their Shapley values, but scales exponentially with dimensionality. Chan & Veas (2024) use a gradient-based approach to rank features based on gradient descent, updating a feature's weight. Wu et al. (2020)'s framework learns global explanations by identifying which feature detectors are most critical for each class, then testing how well those detectors respond to semantic concepts, with Graziani et al. (2020) extending this to medicine, proposing a framework to explain attribute pixel-level values to user-defined concepts. Global Attribution Mapping (GAM) Ibrahim et al. (2019) captures heterogeneous patterns across subpopulations of local explanations clustered on distance metrics to arrive at a singular global explanation. Although faithful, this relies on local attributions computed individually on input data, which is time-consuming and computationally expensive. As the algorithm also computes the distance matrix for all input samples, the time complexity is quadratic in the number of samples, making it unsuitable for large datasets.

## 2.1 KANs for attribution

With KANs being proposed as interpretable networks, there have been applications in medicine. Dong et al. (2025) used a KAN-based Graph Convolutional Networks (GCNs) for AD diagnosis on the ADNI dataset, with their GCN-KAN achieving a classification accuracy of 62.6% over standard GCNs at 57.4%. They score each ROI's importance by summing the absolute spline coefficients from the KAN layer edges connected to that ROI. Although their GCN-KAN identified the hippocampus, inferior parietal gyrus, and amygdala as relevant regions for AD, the approach does not work on voxel-level granularity. Knottenbelt et al. (2025) proposed CoxKAN, a KAN-based framework for interpretable survival analysis with Cox proportional hazards regression, and pruned their network using the L1-norm on the edge-function of their KAN.

KANs have a structure where the flow of information throughout the network can be interpreted and input contributions can be quantified. By tracing the model's decision through the flow of information back to a specific input or set of inputs, it can be made possible to identify and rank the features influencing the model. In Liu et al. (2025), the authors utilize L1 norms to score each layer. The individual L1 norm for each activation $\phi_{i,j}$, defined as $|\phi_{i,j}|_1 = \frac{1}{N_p} \sum_{s=1}^{N_p} \left| \phi_{i,j} \left( x_j^{(s)} \right) \right|$, is calculated, where $\phi_{i,j}$ is the learnable univariate function on the edge from input $j$ to output $i$, $x_j^{(s)}$ denotes the $j$-th feature value from the $s$-th data sample, $N_p$ is the total number of samples.

KAN 2.0 Liu et al. (2024) states that this would not effectively capture the global relationships of the edge with the rest of the network, due to L1 norms capturing just the local information. To solve this, the authors propose a recursive scoring method utilizing standard deviations of activations on the edges and nodes. They consider an $L$-layer KAN with width $[n_0, n_1, \cdots, n_L]$, $E_{l,i,j}$ as the standard deviation of activations on the $(l, i, j)$ edge, and $N_{l,i}$ as the standard deviation on the $(l, i)$ node. They score nodes as $A_{l,i}$ and edges as $B_{l,i,j}$ recursively as follows: all output node scores are set to 1; i.e., $A_{L,i} = 1$ for all $i$. For $l = L, L-1, ..., 1$: $B_{l-1,i,j} = A_{l,j} \frac{E_{l,j,i}}{N_{l+1,j}}$; $A_{l-1,i} = \sum_{j=0}^{n_l} B_{l-1,i,j}$. This iterative computation propagates attribution scores from the output layer to the input layer, capturing the activity of each node. We are looking to score each input with respect to the output, i.e., how much each input contributes to the prediction of the model. We are also looking to find which combinations of edges carry an input's influence to the output, so a path-based attribution is required. Hence, an attribution method that propagates edge scores throughout the network to compute the contribution of each input is necessary.

# 3 Methodology

## 3.1 Edge-Scoring

KANs have a structure in which each edge comprises a function with a linear base term and a nonlinearly modulated spline component, with addition or multiplication at the nodes. A meaningful edge-scoring metric must therefore reflect the influence of this function under data activations. For each edge from input node $i$

to output node $o$, the contribution can depend on both the learned base weight and the projection of the input through a spline basis expansion whose coefficients are learned during training. Several edge-scoring metrics can be used to quantify edge importance in this setting. Examples include the L2 magnitude of the combined base and spline activations, formally defined on edge $E_{i,o}$ as:

$$E_{i,o}^{L2} = \sqrt{\frac{1}{B} \sum_{b=1}^{B} \left( \sum_{c=1}^{C} B_{b,i,c} \, W_{o,i,c}^{\mathrm{spline}} \; + \; W_{o,i}^{\mathrm{base}} \, b(x_{b,i}) \right)^2}, \tag{1}$$

where $B_{b,i,c}$ is the activation of the $c$-th spline basis for input $i$ in sample $b$, $W_{o,i,c}^{\mathrm{spline}}$ is the associated spline weight, $W_{o,i}^{\mathrm{base}}$ is the base weight from $i$ to $o$, and $b(\cdot)$ is the base activation function (e.g. SiLU). Several other metrics can also be used to compute the edge-score, such as the L1 norm Liu et al. (2025), standard-deviation, maximum of base and spline contributions, among several others, defined over the parameters in Eqn. 1. Each of these metrics captures different aspects of edge behavior, such as total activation energy, sparsity, variability, or peak response, and the choice of metric would be dataset-dependent, based on the nature of the data being analyzed.

In our experiments, we evaluate multiple such scoring functions both independently and in combination with the proposed path-based aggregation framework. These choices are not presented as theoretically optimal; rather, they serve as representative instantiations illustrating how different notions of edge importance interact with path-level attribution. Importantly, the proposed framework can be used in tandem with any suitable alternative edge-scoring function that captures the contribution of the edge-function and can be readily applied depending on the dataset, architecture, or analysis objective.

### 3.2 Path-based Attribution

The formulae in Sec. 3.1 capture just the local influence of the function on the edge, and hence, do not account for the contributions of edges downstream in the network. Even if the edge has a strong local influence, it can be cancelled out by an edge later on in the network Liu et al. (2025). To account for the behaviourdownstream contribution of edges towards scoring the contribution of the input, we propose a path-based attribution method. To prevent an explosion of the edge-score, we normalize the score with the tanh function, with motivation from Liu et al. (2025). We score the paths as the absolute of the tanh of the product of edge-scores on the path. The input is scored as a sum of all paths from the respective input node to the output node. If there are several output nodes, all the scores from the specific input node to all the output nodes are averaged. Formally, the importance score of an input feature $i$ in an $L$-layer Kolmogorov-Arnold Network (KAN) with $n$ output nodes can be expressed as the average over all output neurons of the sum over all possible paths $\mathcal{P}_{i \to o}$ from input $i$ to output neuron $o$:

$$\mathrm{imp}_i = \frac{1}{n} \sum_{o=1}^{n} \sum_{p \in \mathcal{P}_{i \to o}} \prod_{e \in p} |\alpha_e|, \text{ where } \alpha_e = \tanh(E_e). \tag{2}$$

Here, the product is taken over all edges $e$ along the path $p$ from input neuron $i$ to output neuron $o$, and $E_e$ is the edge-score at edge $e$. For multiclass settings, the importance scores $\mathrm{imp}_i$ can be obtained by averaging contributions across class-specific output neurons $o$, as in Eqn. 2. While the proposed path-based attribution framework is agnostic to the choice of local edge-scoring function, our experimental evaluation considers multiple edge-scoring metrics as described in Sec. 3.1. These include norm-based, variability-based, and max-based measures, all of which are compatible with the proposed framework and are assessed empirically across datasets. For reporting and visualization, we use magnitude-based (unsigned) edge scores to provide a consistent measure of how strongly an edge is utilized under data-driven activations. Importantly, none of the evaluated edge-scoring metrics is presented as theoretically optimal or universally superior. Instead, the choice of edge-scoring function is treated as dataset-dependent, and our results illustrate how different scoring rules interact with path-level aggregation under varying data characteristics. The proposed framework remains agnostic to this choice and can accommodate suitably alternative metrics without modification.

The non-linear functions in a KAN lie on its respective edges, with simple addition or multiplication performed at the nodes. This means the influence of an input variable factorizes naturally along directed paths

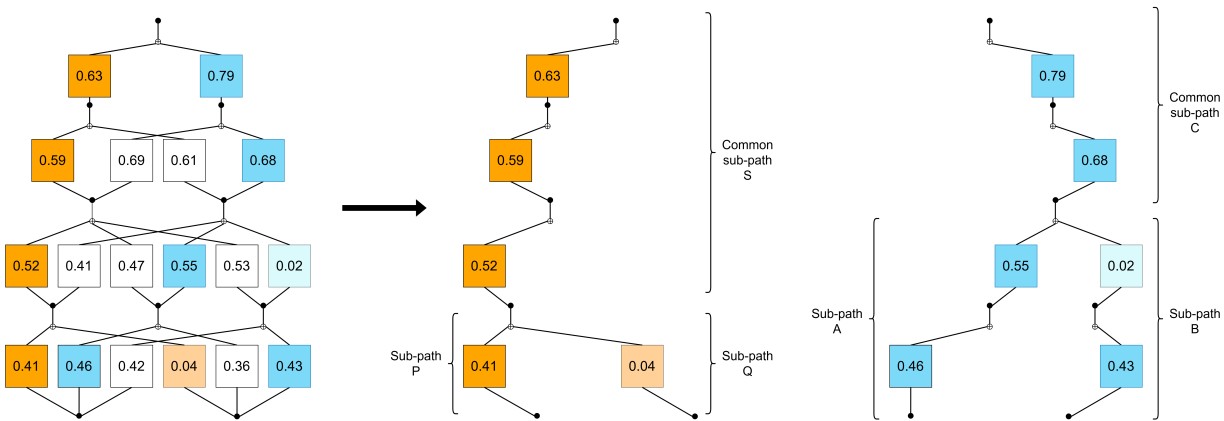

Figure 1: The proposed path factorization avoids double counting in KAN attribution. From left to right: Flow of information is bottom-to-top; A multi-layer KAN showing two different sets of subpaths (orange and blue) to the output node, with example function scores on edges. Decomposition shows how shared subpath S factors identically into both paths, with distinguishing contributions arising only from unique sub-paths A and B; P and Q. The multiplicative factorization ensures that overlapping edges contribute uniformly without additive redundancy, preserving the relative importance of each input.

from itself to the output. While KAN computation graphs are not path disjoint, different inputs often traverse overlapping subnetworks due to shared spline edges. Under these architectural assumptions, where each edge applies a learned function to its input, path-disjoint and each node sums incoming signals, the network output is expressed as a path expansion:

$$y = \sum_i x_i \sum_{P \in \mathcal{P}(i \to o)} \prod_{e \in P} \alpha_e. \tag{3}$$

Consider the illustrative toy-example in Fig. 1. For two paths $P$ and $Q$ that share a common subpath $S$, the factorization gives:

$$\prod_{e \in P} \alpha_e = \left( \prod_{e \in S} \alpha_e \right) \left( \prod_{e \in P \setminus S} \alpha_e \right), \qquad \prod_{e \in Q} \alpha_e = \left( \prod_{e \in S} \alpha_e \right) \left( \prod_{e \in Q \setminus S} \alpha_e \right) \tag{4}$$

where the shared product $\prod_{e \in S} \alpha_e$ is factored identically in both paths as a common multiplicative scalar rather than accumulating additively. As a result, the distinguishing contribution of each path arises only from the edges unique to that path, i.e. from $\prod_{e \in P \setminus S} \alpha_e$ versus $\prod_{e \in Q \setminus S} \alpha_e$. Overlaps are therefore factored in uniformly, propagating the same multiplicative weight to every traversing path, avoiding bias due to double-counting. The measure is non-negative and bounded, producing overlap-neutral attribution that is consistent with the multilayer KAN architectures used in our experiments. The path scoring accounts for downstream effects of the edge on the rest of the network, where any cancellation upstream results in lower relevance, activity and weights downstream. This is similar to relevance propagation in attribution methods such as DeepLIFT Shrikumar et al. (2017) and Layer-wise Relevance Propagation Bach et al. (2015). The input score is then converted to percentage-wise contribution to highlight individual contributions of each input node. This level of granularity is useful for domains such as medical imaging, where regional interpretability is desired. Although the path-based formulation in Eqn. (2) captures both local edge influence and downstream interactions, computing these scores by explicitly enumerating all input-output paths is not feasible for deep or moderately wide KANs, since the number of possible paths grows exponentially with network depth. Therefore, an efficient propagation mechanism is required to obtain the same path-aggregated importance values without enumerating paths directly. We introduce such an efficient matrix-based computation in the following section.

### 3.3 Efficient Path-based Attribution

For a computationally efficient attribution framework, we compute the same input-score using matrix multiplication. Let $\boldsymbol{\alpha}^{(l)} \in \mathbb{R}^{d_l \times d_{l-1}}$ be the edge importance matrix for layer $l$, where $d_0$ is the input dimensionality and $n$ is the number of output neurons. Then, the propagated importance matrix $\mathbf{M}$ is:

$$\mathbf{M} = \boldsymbol{\alpha}^{(L)} \cdot \boldsymbol{\alpha}^{(L-1)} \cdots \boldsymbol{\alpha}^{(1)} \quad \in \quad \mathbb{R}^{n \times d_0}, \tag{5}$$

The importance of the $i$-th input neuron simplifies to $\text{imp}_i = \frac{1}{n} \sum_{o=1}^{n} \mathbf{M}_{o,i}$. This matrix-based calculation scales linearly with the number of edges, $\mathcal{O}\left(\sum_{l=1}^{L} d_l d_{l-1}\right)$, which is same as chained matrix multiplication compared to the exponential complexity of the naive path enumeration, $\mathcal{O}\left(\prod_{l=1}^{L} d_l\right)$, making it feasible for deeper and wider models with numerous paths. This makes the scoring approach practical for high-dimensional neuroimaging data. For example, a structural MRI volume of size $91 \times 109 \times 91$ yields $d_0 = 902{,}629$ voxels. For a KAN with layer widths $d_1 = 256$, $d_2 = 128$, and $d_3 = 64$, the efficient matrix-based propagation $\mathcal{O}\left(\sum_{l=1}^{L} d_l d_{l-1}\right) = 256 \cdot 902{,}629 + 128 \cdot 256 + 64 \cdot 128 \approx 2.31 \times 10^8$, which is significantly smaller than naive path enumeration $\mathcal{O}\left(\prod_{l=1}^{L} d_l\right) = 902{,}629 \cdot 256 \cdot 128 \cdot 64 \approx 1.89 \times 10^{12}$, an increase of more than four orders of magnitude, making exhaustive path computation infeasible. This comparison highlights the necessity of the efficient matrix-based formulation for voxel-level global attribution in high-dimensional neuroimaging settings. Once the inputs have been attributed and scored, they are then converted to percentage-wise contributions to highlight individual contributions of each input node, at input-level granularity. This resolution is very important in brain imaging due to the requirement of high granularity, such that individual brain regions are highlighted to carry out accurate medical inferences.

## 4 Experimental Setup

The datasets utilized in this study include the volumetric public OASIS-1 dataset Marcus et al. (2007) to evaluate the framework's performance on 3D volumetric data, the middle sagittal slice from the ADNI dataset Jack Jr et al. (2008), selected to analyze the volume on 2D-slice data, and the Mendeley coronal-slice MRI collection Yakkundi (2023). We choose the middle sagittal slice as it passes through key midline brain structures (e.g., corpus callosum, cingulate gyrus, ventricles), which are highly informative for Alzheimer's pathology, while reducing dimensionality compared to full 3D volumes Odimayo et al. (2024),Hoang et al. (2023). All three datasets comprise sMRI scans of cognitively normal individuals, subjects with mild cognitive impairment (MCI), and patients diagnosed with AD. These cohorts provide a progression of disease states and are established benchmarks for Alzheimer's prediction on neural networks. OASIS consists of 436 volumes of shape $91 \times 109 \times 91$. ADNI consists of 592 volumes, from which we extract the sagittal slice and crop it to $184 \times 152$. The Mendeley data cohort consists of 6400 samples of coronal slices. Class imbalance is handled by the usage of weighted Cross-Entropy loss. Standard preprocessing procedures such as skull-stripping Smith (2002)Smith et al. (2004), intensity normalization Tustison et al. (2010), and affine registration Avants et al. (2009), were applied to ensure anatomical consistency and comparability across scans. We use these datasets with well-defined diagnostic categories to rigorously test how KANs capture structural differences associated with cognitive decline. As a Proof-of-Concept, we also test the attribution frameworks on a 10-class MNIST classification.

All experiments were conducted on a machine with Nvidia A100 GPU, and the path-based attribution framework is applied post-hoc on fully trained models to generate global attribution maps, showing the contribution of specific inputs towards the models decision. The maps are quantitatively assessed using Insertion AUC (InsAUC), Deletion AUC (DelAUC) Petsiuk et al. (2018), and Sufficiency (Suff) metrics DeYoung et al. (2020). InsAUC measures how the model's confidence increases as the most important features (voxels) identified by the attribution are progressively added to a blank input. DelAUC assesses how quickly the model's confidence decreases as these important features are removed, indicating the extent to which the model relies on these features for its decision. Suff measures whether the top-ranked features alone are sufficient to maintain the model's predictive performance. This allows for an assessment of whether the network identifies clinically meaningful features for differentiating between healthy and diseased individuals.

Further, to interpret model attributions and assess their relevance, we performed region-of-interest (ROI) analyses on the attribution maps generated across 15 independent runs. Two anatomical atlases were used for comprehensive grey and white matter coverage: the Brainnetome (BN) Atlas Fan et al. (2016) and the Johns Hopkins University (JHU) ICBM-DTI White Matter Atlas Mori et al. (2006). Please refer Appendix A.1 for more details on model setup, training and analysis.

## 5    Results and Discussions

We train a KAN on the datasets and achieve test accuracies of 93.24%, 81.85% and 91.25% respectively, on the OASIS-1, ADNI and Mendeley test sets. We also train the KAN on MNIST classification as a Proof-of-Concept (PoC) and achieve a test accuracy of 99.4%. Using the proposed path-based attribution, we obtain an Insertion AUC of $0.865 \pm 0.049$, Deletion AUC of $0.282 \pm 0.020$, and Sufficiency of $0.068 \pm 0.064$ on the OASIS dataset as indicated in Table 1. On ADNI, the method achieves an Insertion AUC of $0.758 \pm 0.044$, Deletion AUC of $0.380 \pm 0.051$, and Sufficiency of $0.209 \pm 0.143$. For the Mendeley cohort, we report an Insertion AUC of $0.791 \pm 0.016$, Deletion AUC of $0.463 \pm 0.099$, and Sufficiency of $0.226 \pm 0.033$. On MNIST, the results are Insertion AUC $0.964 \pm 0.003$, Deletion AUC $0.393 \pm 0.022$, and Sufficiency $0.072 \pm 0.015$. These values indicate that path attributions consistently rank features that affect model confidence under both insertion and deletion perturbations. We compare against global attribution methods from GAM, KAN, and KAN 2.0, while also reporting results for the proposed path-attribution framework paired with selected edge-scoring functions to illustrate its behaviour, under different scoring choices. It is to be noted that, although this framework has been applied for the pathology of Alzheimer's in this study on 3D-volumetric and 2D-slice data, it can be applied to other neurological conditions and problems involving interpreting KANs where global, population-level pattern identification across subjects and data is required.

### 5.1    Comparative Analysis

Table 1 compares the proposed path-based attribution framework against representative global attribution and edge-scoring approaches derived from GAM, KAN, and KAN 2.0. For GAM, we adopt local attribution scoring via Integrated Gradients (IG), which performed better than simple gradients Ibrahim et al. (2019). We include the L2 norm (Eq. 1), and the maximum of the base and spline contributions of the KAN edge function as two examples of edge-scoring metrics, denoted as $Path\text{--}Attr. + L2$ and $Path\text{--}Attr. + max$. These choices serve as representative examples of edge-scoring strategies within the proposed framework. Statistical significance was evaluated using a two-sided paired Wilcoxon signed-rank test over 15 random seeds, applied independently for each dataset, metric, and method comparison. For the proposed framework, we additionally report, for each dataset–metric pair, the best-performing path-attribution variant to reflect the framework's upper-bound capability under the considered scoring functions. It can be observed that the path-attribution framework achieves statistically significant improvements over baseline methods on Mendeley and on MNIST, where the task is comparatively simple and base model accuracy is high. In contrast, performance on Insertion AUC against GAM is statistically indistinguishable on OASIS and ADNI, as are Deletion AUC comparisons against KAN and Sufficiency comparisons against GAM on OASIS and KAN on ADNI, other than which, the proposed framework is significantly better than baselines. Within the proposed framework itself, quantitative differences across edge-scoring norms are also observed: L2 attains the highest path-attribution Insertion AUC on all datasets (OASIS: 0.865, ADNI: 0.758, Mendeley: 0.791, MNIST: 0.964), while competing norms produce consistently lower insertion scores by margins of 0.009–0.028, suggesting that, for these datasets, attribution mass accumulated under L2 leads to larger increases in prediction confidence during feature insertion. For Deletion AUC, the minimum values shift across datasets: L2 on OASIS, L1 on ADNI and Mendeley, and STD on MNIST, suggesting dataset-dependent differences in how rapidly model confidence decreases under feature removal. Similarly, for Sufficiency, the lowest scores are achieved by STD on OASIS, MAX on ADNI, and L2 on Mendeley and MNIST, suggesting differences across datasets in the proportion of features required to retain predictive performance. Overall, the results indicate that the advantages of the proposed path-attribution framework are dataset- and metric-dependent, with clearer improvements emerging in settings where attribution signals are more pronounced. In cases where differences are statistically insignificant, the results suggest comparable explanatory performance between methods rather than definitive superiority of any single approach.

Nevertheless, the results in Table 1 demonstrate that path-based multilayer attributions provide competitive and often improved performance relative to established attribution baselines, without observable degradation across the evaluated settings. These findings support the use of path-based attribution as an interpretability tool in both neuroimaging-focused and general prediction tasks, contingent on adequate underlying model performance.

Table 1: Comparison of Path-Attribution variants against baselines. Included are two example edge-scoring metrics along with the best performing edge-scoring metric in tandem with the proposed framework.

| Dataset (Accuracy, AUC) | Method | InsAUC ↑ (± std) | DelAUC ↓ (± std) | Suff ↓ (± std) |
|---|---|---|---|---|
| OASIS (Acc: 93.24%, AUC: 95.74) | Path-Attr. (best) | l2: 0.865 ± 0.049 | l2: 0.282 ± 0.020 | std: 0.042 ± 0.068 |
| | Path-Attr. + l2 | 0.865 ± 0.049 | 0.282 ± 0.020 | 0.068 ± 0.064 |
| | Path-Attr. + max | 0.837 ± 0.015 | 0.331 ± 0.080 | 0.103 ± 0.076 |
| | GAM + IG | 0.860 ± 0.048 | 0.366 ± 0.161 | 0.046 ± 0.071 |
| | KAN 2.0 | 0.845 ± 0.030 | 0.354 ± 0.101 | 0.087 ± 0.073 |
| | KAN | 0.839 ± 0.042 | 0.287 ± 0.024 | 0.072 ± 0.058 |
| ADNI (Acc: 81.85%, AUC: 91.30) | Path-Attr. (best) | l2: 0.758 ± 0.044 | **l1: 0.310 ± 0.081** | max: 0.201 ± 0.134 |
| | Path-Attr. + l2 | 0.758 ± 0.044 | 0.380 ± 0.051 | 0.209 ± 0.143 |
| | Path-Attr. + max | 0.749 ± 0.059 | 0.364 ± 0.041 | 0.201 ± 0.134 |
| | GAM + IG | 0.746 ± 0.026 | 0.506 ± 0.048 | 0.269 ± 0.136 |
| | KAN 2.0 | 0.716 ± 0.033 | 0.439 ± 0.034 | 0.326 ± 0.124 |
| | KAN | 0.666 ± 0.016 | 0.376 ± 0.075 | 0.221 ± 0.029 |
| Mendeley (Acc: 91.25%, AUC: 96.62) | Path-Attr. (best) | **l2: 0.791 ± 0.016** | **l1: 0.296 ± 0.053** | **l2: 0.226 ± 0.033** |
| | Path-Attr. + l2 | 0.791 ± 0.016 | 0.463 ± 0.099 | 0.226 ± 0.033 |
| | Path-Attr. + max | 0.773 ± 0.012 | 0.513 ± 0.067 | 0.265 ± 0.093 |
| | GAM + IG | 0.767 ± 0.037 | 0.399 ± 0.090 | 0.293 ± 0.113 |
| | KAN 2.0 | 0.781 ± 0.019 | 0.554 ± 0.027 | 0.301 ± 0.034 |
| | KAN | 0.710 ± 0.080 | 0.401 ± 0.053 | 0.444 ± 0.102 |
| MNIST (Acc: 99.41%, AUC: 99.89) | Path-Attr. (best) | **l2: 0.964 ± 0.003** | **std: 0.114 ± 0.007** | **l2: 0.052 ± 0.015** |
| | Path-Attr. + l2 | 0.964 ± 0.003 | 0.293 ± 0.022 | 0.052 ± 0.015 |
| | Path-Attr. + max | 0.944 ± 0.004 | 0.269 ± 0.009 | 0.161 ± 0.015 |
| | GAM + IG | 0.956 ± 0.004 | 0.380 ± 0.008 | 0.103 ± 0.013 |
| | KAN 2.0 | 0.911 ± 0.010 | 0.326 ± 0.111 | 0.412 ± 0.164 |
| | KAN | 0.924 ± 0.012 | 0.240 ± 0.012 | 0.111 ± 0.015 |

**Statistical Significance:** $p < 0.05$ | $p < 0.01$ | $p \geq 0.05$

## 5.2 Visual Comparisons

The side-by-side heatmaps of the attribution maps are presented in Table 2, where the middle slice is plotted along the sagittal axis for OASIS and ADNI, and the coronal slice is presented for the Mendeley Dataset, along with the MNIST global attribution maps. The KAN, KAN 2.0, and GAM plots are presented alongside the proposed path-based attribution approach. Although the general regions identified by all approaches appear to be similar, there are differences in the magnitude and number of regions highlighted. The KAN approach has a lot of area highlighted as downstream path interactions are not considered. KAN 2.0 sees some regions that are highlighted by our approach and GAM as not highlighted, as some functions that contribute equally without a large standard deviation are suppressed.

It can be observed in the ADNI maps that the slice provides a visualization of midline structures such as the corpus callosum, cingulate gyrus, thalamic region, and the ventricular cavity, all of which exhibit different tissue intensity patterns. Several of these structures are known to exhibit degeneration in AD. The smooth high-intensity band near the center corresponds to dense white-matter tracts of the corpus callosum

Walterfang et al. (2014), while surrounding regions with moderate intensity reflect gray-matter cortical areas. The darker cavities near the center and lower portions of the image represent cerebrospinal fluid (CSF) spaces, primarily the lateral and third ventricles. Even though volumetric and ROI-level analyses are not feasible on this single slice, it remains structurally informative due to the inclusion of midline anatomy that captures the general morphological organization of the brain.

The Mendeley attributions show the coronal mid-slice showing pronounced activations along the medial temporal regions, particularly surrounding the hippocampal and parahippocampal areas Echávarri et al. (2011), and extending around the periventricular zones. Highlighted regions are also visible across the cortical ribbon, outlining the gyral-sulcal boundaries along the outer cerebrum. The central midline region-corresponding primarily to the thalamus, corpus callosum, and basal ganglia structures-appears comparatively subdued, with minimal activation intensity. This lower response suggests that the discriminative emphasis in this slice arises more from cortical and medial temporal morphometry than from deep gray matter contrast.

Table 2: Visual comparisons with example attribution maps

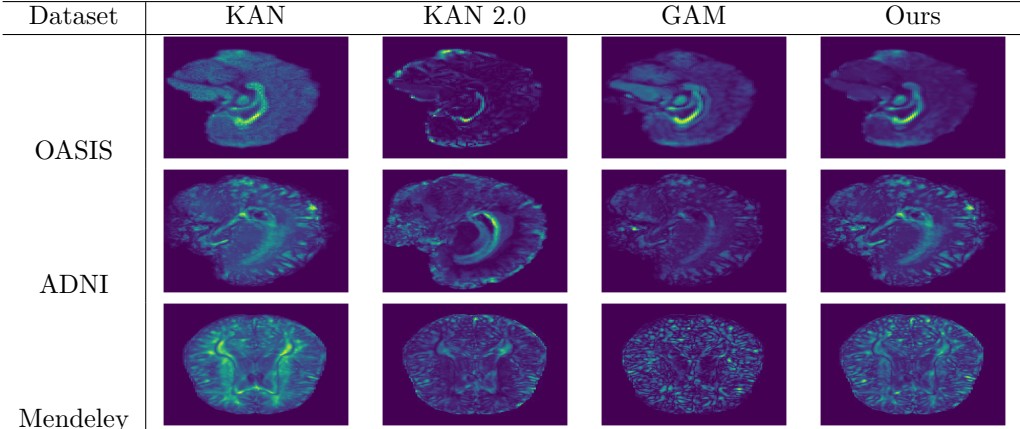

| Dataset | KAN | KAN 2.0 | GAM | Ours |
|---------|-----|---------|-----|------|
| OASIS | | | | |
| ADNI | | | | |
| Mendeley | | | | |

### 5.2.1 Medical Insights from the OASIS Attribution Maps

The mapped ROI inferences on the OASIS volume on grey matter from the Brainnetome atlas were as follows. The bilateral dorsal caudate nuclei (dCa_R and dCa_L) were the most important region pair observed in Fig. 2c. Prior studies have reported caudate nucleus atrophy in AD, supporting its involvement in disease progression Jiji et al. (2013). The regions have mean attribution importance of 0.759±0.144 (Coefficient of Variation (CV) = 18.91%) and 0.687±0.109 (CV = 15.87%) respectively. As seen in Fig. 2, the bilateral medial prefrontal cortex (A10m_R: 0.571±0.016, CV = 2.76%; A10m_L: 0.568±0.019, CV = 3.39%), left angular gyrus (A39rv_L: 0.487±0.026, CV = 5.38%), and left cuneus (cCunG_L: 0.458±0.031, CV = 6.85%) all show perfect consistency. The medial prefrontal regions showed the highest stability, and convergence of default mode network regions with frontostriatal structures is consistent with established AD neurobiology. Bilateral superior temporal regions (TE1.0/TE1.2_L: 0.672±0.063; TE1.0/TE1.2_R: 0.531±0.040) also were fully consistent, indicating temporal lobe involvement in AD Castellano et al. (2024). Near-perfect consistency was observed for the right inferior parietal cortex (A40c_R: 0.461±0.025) The right hippocampus (cHipp_R) appeared in 66.7% of runs with high stability when present (CV = 2.90%, range 0.433-0.476). The top grey matter ROIs comprised prefrontal cortex (4 ROIs: bilateral A10m, A10l_R, A13_L), subcortical regions (2 ROIs: bilateral caudate), parietal cortex (4 ROIs: A39rv_L, A40c_R/L, A40rv_R), temporal cortex (2 ROIs: bilateral TE1.0/TE1.2), posterior cortex (1 ROI: cCunG_L), and medial temporal lobe (1 ROI: cHipp_R). Independent OASIS studies identified overlapping anatomical regions with our findings: Castellano et al. (2024) identified temporal and frontal regions as critical for classification; Kim et al. (2024) identified visual cortices, caudate and hippocampus as AD contributors.

The top white matter ROIs identified from the JHU atlas were as follows. The body of corpus callosum (importance = 2.331) and genu of corpus callosum (importance = 2.167) emerged as the highest-ranked commissural structures, as seen in Fig. 2b. Corpus callosum atrophy in AD reflects interhemispheric

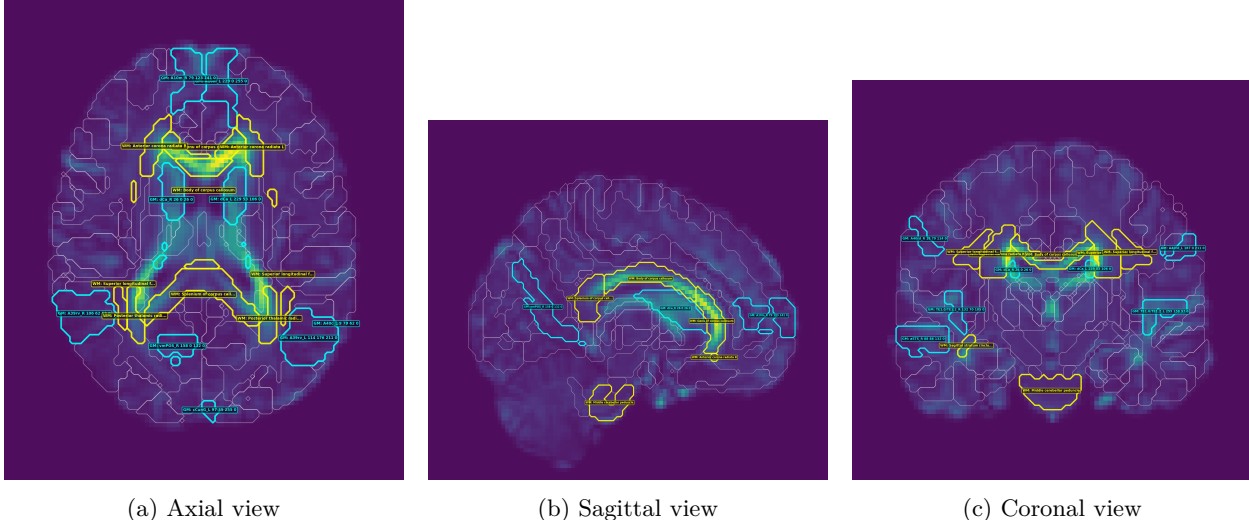

(a) Axial view         (b) Sagittal view         (c) Coronal view

Figure 2: Top ROIs identified through attribution analysis in (a) axial, (b) sagittal, and (c) coronal views, showing gray matter (GM) and white matter (WM) regions most contributive to model predictions. The regions primarily highlighted in the maps were body and genu of the Corpus Callosum, Bilateral posterior thalamic radiation, the prefrontal, parietal, temporal and posterior cortex

commissural fiber degeneration secondary to cortical neuronal loss Tomimoto et al. (2004). The splenium of corpus callosum (importance = 1.236) showed third-highest importance; posterior callosal regions show particular vulnerability in AD due to their connections with temporal and parietal cortices Kamal et al. (2021). Association fiber tracts demonstrated substantial importance scores. Bilateral posterior thalamic radiation (left: 0.866, right: 0.847) ranked among the top white matter regions of interest, consistent with established involvement of thalamic relay structures in AD populations Biesbroek et al. (2024). Bilateral anterior corona radiata (left: 0.848, right: 0.762) and bilateral superior corona radiata (left: 0.709, right: 0.632) were also identified among the ROIs seen in Fig. 2a. The corona radiata comprises projection fibers connecting cortical regions through the internal capsule to subcortical structures; compromised white matter within corona radiata tracts has been reported in AD population. The middle cerebellar peduncle (importance = 0.660) emerged with moderate importance, reflecting potential cerebellar involvement in cognitive processing deficits. Bilateral posterior corona radiata (left: 0.546, right: 0.531) completed the set of significant white matter tracts identified. Collectively, these white matter tract findings reflect regions that are previously reported to be affected in AD. This convergence across highlighted grey and white matter ROIs suggests the biological relevance and plausibility of our attributed regions with respect to established AD pathology.

### 5.3 Limitations

To analyze the performance of our framework on 2D inputs, the ADNI experiments utilize only the middle sagittal slice rather than full 3D volumes. A limitation of the proposed framework is its dependence on model performance; attribution maps are only meaningful to the extent that the underlying model has learned a reliable decision function, which is 93.24%, 81.85% and 91.25% accurate on the OASIS, ADNI and Mendeley datasets. The framework currently does not provide inherently faithful class-conditional global attribution maps in fully connected KANs. Because edge functions are learned jointly across the full training distribution, and we analyze just the trained model without relying on input-data post-hoc, restricting path aggregation to individual output neurons does not guarantee disentangled or class-specific explanations. As a result, the attribution maps produced are global and cumulative, reflecting the complete learned behavior of the model rather than class-wise patterns. The framework is also sensitive to architectural choices, such as network depth and width, the choice of edge-scoring function, and to the quality of the trained model. In

particular, the method is most effective in deeper networks where meaningful information flow can be traced across multiple layers.

## 6 Broader Impact Statement

The proposed path-based attribution framework is intended as a research and decision-support tool for understanding population-level information flow in trained models, rather than as a standalone diagnostic system. In medical imaging contexts, a primary ethical concern is the risk of automation bias, where clinicians or practitioners may place undue trust in attribution maps and interpret them as ground-truth biomarkers. Such over-reliance could lead to inappropriate clinical conclusions, particularly if the underlying model has learned spurious correlations or dataset-specific artifacts. Accordingly, the attribution maps produced by this framework should be interpreted as suggestive explanations that support hypothesis generation, model auditing, and exploratory analysis, and not as definitive evidence for clinical decision-making. Any clinical use must occur in conjunction with established diagnostic protocols and expert judgment. Furthermore, this study relies on widely used public neuroimaging datasets, including OASIS, ADNI, and Mendeley, which are known to exhibit demographic and socioeconomic biases and to underrepresent certain populations. Because the proposed method derives global attributions from the training distribution, the resulting maps may underrepresent or mischaracterize patterns present in underrepresented demographic groups. As a result, conclusions drawn from these attribution maps should be contextualized with respect to the populations on which the models were trained. Validation on more diverse, real-world cohorts is necessary before considering any clinical translation. Finally, global attribution methods, including the one proposed in this work, do not eliminate the need for careful dataset curation, robust model evaluation, and interdisciplinary collaboration. When used appropriately, the framework can aid researchers in identifying systematic model dependencies and potential biases, but attribution maps should be interpreted carefully, while also considering the broader context of clinical and scientific evidence.

## 7 Conclusion

We introduced a path-based attribution framework for Kolmogorov-Arnold Networks that generates global, population-level importance maps by tracing information flow through the network's explicit edge structure. The proposed framework produces faithful attribution maps that reflect intrinsic network properties learned during training. The efficient matrix-based implementation scales linearly with network depth, making the approach practical for high-dimensional neuroimaging applications. Our evaluation across three publicly available AD classification datasets (OASIS, ADNI, Mendeley) and MNIST shows that the proposed method consistently performs competitively against existing global attribution approaches in attribution metrics. The attribution maps identify clinically meaningful brain regions including the bilateral caudate nuclei, medial prefrontal cortex, and default mode network structures, with high consistency across multiple runs and concordance with established AD neurobiology literature. Higher Insertion AUC and lower Deletion AUC values also show that our method accurately identifies the most informative features for model predictions.

This work addresses a present gap in medical AI interpretability by providing voxel-level global attributions for KAN-based neuroimaging models. The ability to generate population-level importance maps as network-intrinsic properties supports the identification of biomarkers across patient populations, assisting clinical validation and regulatory approval of AI systems. By bridging the gap between model interpretability and medical knowledge, our framework can facilitate trust and transparency in AI-assisted diagnosis.

Several promising directions emerge from this work. Extending the framework to multi-class classification with ways to generate disentangled class-wise maps would broaden its applicability. Applying the method to other medical imaging modalities and domains beyond medicine, such as on tabular data, would validate its generalizability. Exploring ways to generate classwise attribution maps and bettering KAN performance on a larger scale data is one of our ongoing works. Finally, developing interactive visualization tools that allow clinicians and users to explore path-level attributions could enhance trust in KAN-based systems.

## Acknowledgements

The authors would like to thank the anonymous reviewers and the Action Editor for their valuable feedback, which has significantly improved the quality of this manuscript. We are grateful to Priyam Dey for his helpful input towards revision of the manuscript, and K S Suryanarayan for his contribution towards Fig. 1. This work was also partially supported by EMSTAR (Pratiksha Trust Extra-Mural Support for Transformational Aging Brain Research).

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

# A Appendix

## A.1 Training and Implementation Details for Models and Metrics

All experiments were carried out using the Kolmogorov–Arnold Network with weight decay architecture and trained on a CUDA-A100. Dependencies necessary for execution of the framework included PyTorch 2.5.1+cu121 as the primary deep learning framework. Supporting data processing and analysis were performed using NumPy 1.26.4, Pandas 2.2.2, and scikit-learn 1.2.2. Neuroimaging data was handled using NiBabel 5.3.2. Each model was built using a sequence of `KANLinear` layers, where the connection weights are represented using B-spline functions. Unless stated otherwise, the following spline and training settings were kept the same across all datasets:

- grid size: 5 points per input dimension,

- spline order: cubic (order 3),

- base activation function: SiLU,

- noise scale: 0.1,

- linear and spline scaling factors: 1.0,

- grid epsilon: 0.02,

- initial spline grid range: $[-1, 1]$.

**Spline configuration and initialization.** All KAN models use cubic B-spline basis functions with five knots per input dimension, following the configuration recommended in the original KAN implementation Liu et al. (2025). This choice was kept fixed across all experiments to avoid introducing additional hyper-parameter variation. Although this setting provided stable training and sufficient expressivity for the tasks considered in this work, we do not claim that five knots are universally optimal.

**Initialization noise.** The noise scale refers to the standard deviation of a small additive Gaussian noise applied during spline coefficient initialization. This initialization strategy, adopted from the original KAN implementation, encourages smoother function learning and helps prevent degenerate spline configurations early in training.

**Spline grid and scaling across layers.** Spline grids are defined over the interval $[-1, 1]$, and input features are scaled accordingly before being passed to the first KAN layer. Subsequent layers operate on the activations produced by preceding KAN layers; these activations remain within the effective spline support due to the learned base and spline functions, and therefore do not require additional rescaling.

The layer widths were selected to reflect the input dimensionality and complexity of each dataset. For OASIS-1, the network width configuration was

$$[D, 20, 10, 10, 10, 10, 10, 10, 10, 5, 1],$$

where $D$ denotes the number of input features. This produced an 11-layer network with progressively narrower layers for binary classification.

For ADNI, a larger first hidden layer was used to accommodate its higher input dimensionality. The width configuration was

$$[D, 800, 20, 15, 15, 15, 10, 10, 10, 5, 1].$$

For the Mendeley dataset, a slightly shallower network was used:

$$[D, 800, 20, 15, 15, 10, 10, 10, 1].$$

All models were optimized using AdamW with a learning rate of $10^{-3}$. An exponential learning rate scheduler was applied by a factor of 0.8. The same optimizer and scheduler settings were used for all datasets for reproducibility.

These configurations were sufficient for stable training across datasets and provided a controlled setting for comparing model behaviour. All attribution analyses are performed post hoc on the fully trained models.

**Quantitative Analysis Metrics** To assess and compare our methods against other attribution methods that capture importance across training data, we utilized Insertion and Deletion AUC metrics, with their formal definitons as follows.

Let $f(\cdot)$ denote a trained model, $x \in \mathbb{R}^d$ an input, and $A(x)$ an attribution map that induces a ranking of input features. Let $S_k \subset \{1, \ldots, d\}$ denote the set of top-$k$ features according to this ranking. We define $x_{S_k}$ as the input where only features in $S_k$ are retained and all others are replaced by a baseline value, and $x_{\setminus S_k}$ as the input where features in $S_k$ are removed and replaced by the baseline. The **Insertion AUC** is defined as the area under the curve obtained by evaluating $f(x_{S_k})$ as $k$ increases from 0 to $d$. The **Deletion AUC** is defined as the area under the curve obtained by progressively removing the top-ranked features from the input consisting the full set of features. Formally, we evaluate $f(x_{\setminus S_k})$ for $k = 0, 1, \ldots, d$. The **Sufficiency** metric measures whether the most relevant features alone are sufficient to preserve the model prediction, and is defined as $\text{Suff}(k) = f(x) - f(x_{S_k})$. As a Proof-of-Concept, we also test the attribution frameworks on 10-class MNIST classification.

**Atlases for ROI Analysis** To interpret model attributions and further assess their stability, we performed region-of-interest (ROI) analyses on the attribution maps generated across 15 independent runs. Two anatomical atlases were used for comprehensive grey and white matter coverage: the Brainnetome (BN) Atlas Fan et al. (2016) and the Johns Hopkins University (JHU) ICBM-DTI White Matter Atlas Mori et al. (2006).

The BN Atlas provides a 246-region parcellation of the grey matter at a 2 mm isotropic resolution. It contains both structural and functional connectivity information to define connectivity-informed cortical and subcortical ROIs. This atlas enables detailed inference across diverse grey matter regions and aligns well with OASIS T1-weighted MRI volumes ($91 \times 109 \times 91$ voxels) after standard MNI registration. ROI-level intensity measures and region-based features were extracted for each subject in native space following co-registration.

The JHU ICBM-DTI Atlas offers a 48-region parcellation of the brain's white matter based on high-quality diffusion tensor imaging acquired in the ICBM-152 template space, also at a 2 mm isotropic resolution. Each region corresponds to a major white matter tract manually delineated using fiber orientation information, ensuring standardized tract identification across subjects. For every OASIS MRI, the JHU atlas was co-registered to native space after MNI transformation to extract regional white matter features. This facilitates tract-based ROI inferences for comparative and cross-sectional evaluation of attribution stability within white matter structures.

Together, the BN and JHU atlases provide complementary insights into grey and white matter attribution patterns, allowing for a comprehensive region-level interpretation of the model's behavior across both tissue domains and insights into regions affected by Alzheimer's.

**Usage of the Path-Attribution Framework.** The proposed path-attribution framework is applied post hoc to a fully trained feedforward network that satisfies the following architectural assumptions: (i) nonlinear transformations are associated with edges, (ii) nodes aggregate incoming signals additively, and (iii) directed paths exist from input features to output neurons. The method requires access only to the trained model parameters, including learned edge functions and base activations, and optionally stored activation statistics used for edge scoring. Access to the original training data is not required. Given such a model, attribution is performed by first computing a scalar importance score for each edge using a predefined edge-scoring function, such as the L1, L2, or max norm of the learned function or its activations. These edge scores are then propagated layer-wise using a matrix-based formulation to implicitly accumulate contributions

---

**Algorithm 1:** Path-Based Attribution Framework (Post-hoc)

---

**Input:** A well-trained feedforward network satisfying: (i) nonlinear functions are associated with edges, (ii) nodes aggregate incoming signals additively or multiplicatively, (iii) directed paths exist from input features to output neurons; saved edge functions $\{\phi_{j,k}^{(\ell)}\}$ and trained model parameters.

Let $L$ denote the number of layers, $d_0$ the input dimensionality, $d_\ell$ the width of layer $\ell$, and $n \equiv d_L$ the number of output neurons.

**Output:** Global input-level attribution scores $\mathbf{s} \in \mathbb{R}^{d_0}$

**for** *each layer $\ell = 1, \ldots, L$* **do**

Construct the edge-importance matrix $\alpha^{(\ell)} \in \mathbb{R}^{d_\ell \times d_{\ell-1}}$ with entries

$$\alpha_{j,k}^{(\ell)} \leftarrow \tanh\left(E_{j,k}^{(\ell)}\right), \qquad E_{j,k}^{(\ell)} = g\left(\phi_{j,k}^{(\ell)}\right),$$

where $g(\cdot)$ is a fixed scalar edge-scoring functional;

Compute the propagated importance matrix

$$\mathbf{M} \leftarrow \alpha^{(L)} \cdot \alpha^{(L-1)} \cdots \alpha^{(1)} \in \mathbb{R}^{n \times d_0},$$

which aggregates contributions over all directed paths from inputs to outputs;

**for** *each input feature $i \in \{1, \ldots, d_0\}$* **do**

$$s_i \leftarrow \frac{1}{n} \sum_{o=1}^{n} |\mathbf{M}_{o,i}|$$

Optionally normalize $\mathbf{s}$ for visualization or comparison

---

along all directed paths from input features to output neurons, without explicit path enumeration. Because computation graphs need not be path-disjoint, overlapping paths through shared edges are naturally handled by this formulation.

For each input feature, path contributions are aggregated across all output neurons; in multi-class settings, this aggregation is performed by averaging over outputs to obtain a global, class-agnostic attribution score. The resulting input-level scores may optionally be normalized for stability or visualization. Finally, these scores are mapped back to the input space (e.g., voxels or pixels) to produce a global attribution map summarizing the model's learned information flow across the training distribution.

### A.2 Synthetic Data Test for Proof-of-Concept

To complement evaluations on real-world datasets, we conduct a controlled synthetic experiment designed to verify whether the proposed path-based global attribution framework can correctly recover known signal regions, whilst ignoring background information.

**Dataset construction.** We generate a synthetic dataset consisting of 10,000 grayscale images of size $40 \times 40$. Each image contains a filled triangular shape centered in the frame. For 50% of the samples, a circular/plus-shaped hole is carved out from a fixed subregion located in the lower portion of the triangle, depicted by the red region in Fig. 3c, while the remaining samples contain no hole. Additionally, the 70% of the triangles are stretched and squeezed, to denote background noise. The classification task is binary: distinguishing between triangles with and without a hole, shown in Figs. 3b and 3a. Importantly, the hole's general location is clearly defined across samples, ensuring that the discriminative signal is spatially localized and known by construction. .

**Global-Attribution.** A feedforward KAN is trained on this dataset using standard supervised learning until convergence. After training, we apply the proposed path-based attribution method to compute a global

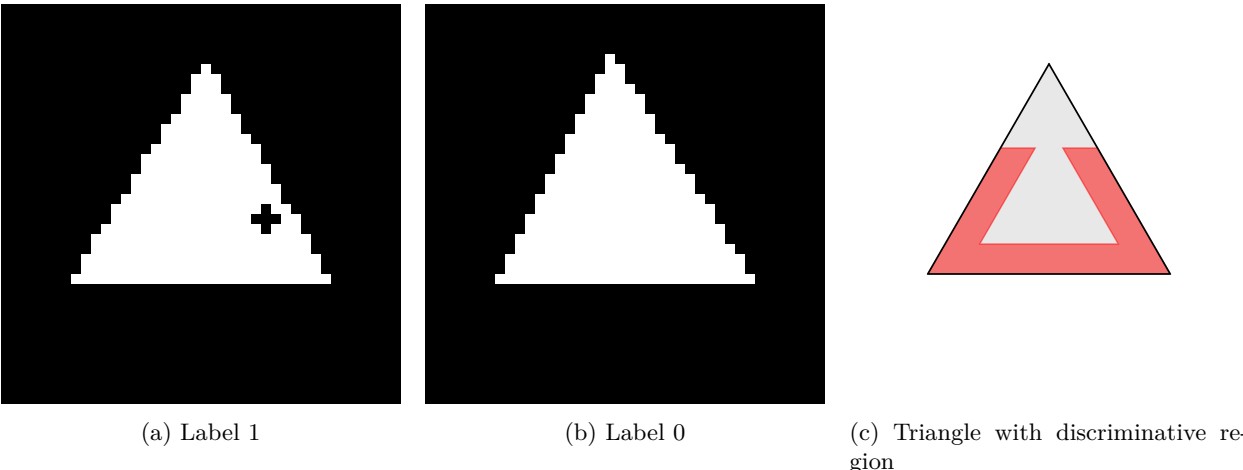

(a) Label 1                     (b) Label 0                     (c) Triangle with discriminative region

Figure 3: Synthetic experiment setup: L-R, Label 1 image with the hole, Label 0 image without a hole; Theoretical triangle, the red region indicates the discriminative area with holes that attribution methods are expected to highlight.

importance map over input pixels. Since the dataset is synthetic and the discriminative signal is explicitly defined, the expected behavior of a faithful global attribution method is clear: the attribution map should highlight the entire subregion of the triangle where holes occur, while assigning low importance to the rest of the image.

**Results and interpretation.** The resulting attribution map accurately localizes the predefined hole-containing subregion and assigns negligible importance to pixels outside this area, including other parts of the triangle that are visually salient but irrelevant to the classification task, as observable in Fig. 4. This demonstrates that the proposed framework captures the true causal structure of the data-generating process, rather than relying on spurious correlations or global shape cues.

**Discussion.** This synthetic experiment serves as a proof-of-concept sanity check, illustrating that the proposed global attribution framework can correctly identify signal features when ground truth is known. While such controlled settings do not reflect the full complexity of real-world data, they provide important validation that the method behaves as intended and that observed attribution patterns on real datasets are not merely artifacts of the evaluation metrics. We emphasize that this experiment is not meant to replace real-data analysis, but to complement it by establishing baseline faithfulness under idealized conditions. To complement this experiment, we also plot the global attribution plot on MNIST as seen in Fig. 5, and we can see the highlighted region focuses on the center, where digits are present, as opposed to the frame. Together with the synthetic experiment, the MNIST plot and the ROI-analysis, we conclude that the proposed framework can identify ground-truth regions effectively.

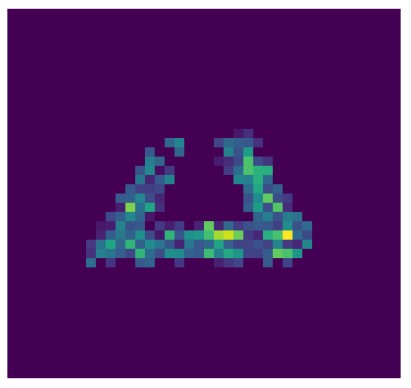 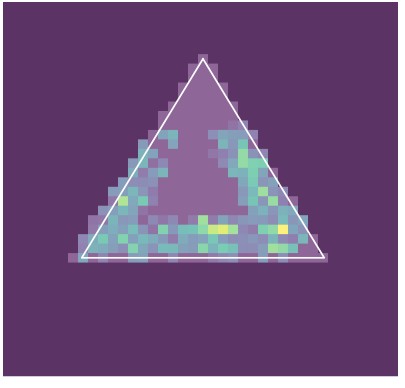 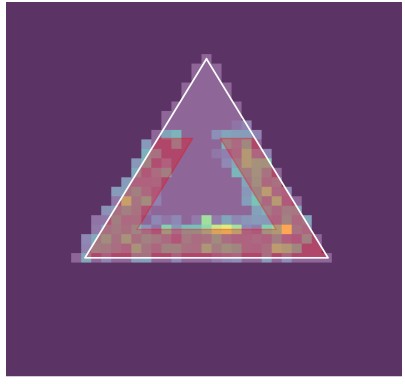

(a) Attributed region generated by the path-attribution method

(b) An overlay of the theoretical triangle over the actual triangle overlaid on the attributed region

(c) An overlay of the theoretical triangle with the discriminative region over the actual triangle overlaid on the attributed region

Figure 4: L-R: The generated attribution map, Overlaid regular triangle region without boundary distortion (not-noise) perfectly encompassing the highlighted region, Theoretical region and actual region

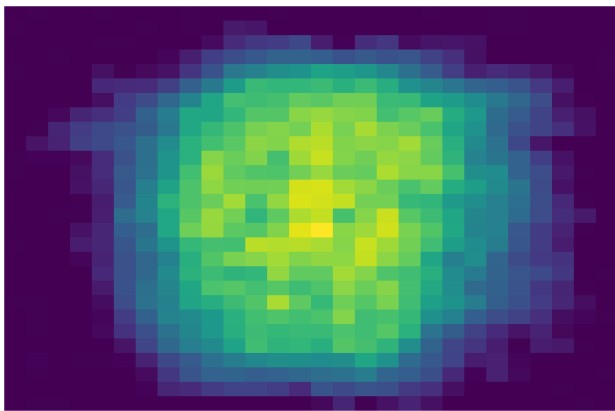

Figure 5: Global-Attribution Map (Path-Attribution + L2 edge scoring) for the MNIST dataset, the highlighted region focuses on the center, where digits are present, as opposed to the frame.

### A.3 Detailed Analysis Table

Comprehensive comparison of explanation quality metrics across four datasets (OASIS-1, ADNI, Mendeley, and MNIST), evaluating 5 attribution methods applied to KAN and Path-Attr. models. Metrics include Insertion/Deletion AUC (InsAUC/DelAUC), and Sufficiency (Suff). Best performance per dataset and metric appears in bold, if there is no significantly better metric, none are bolded. The base KAN paper uses L1 norms at the input node, and we test it with different scoring functions, listed as KAN + scoring function. Rather than claiming a universally optimal scoring rule, our results empirically compare different edge-scoring functions on their own, in-tandem with Path-Attribution and against baselines KAN, KAN 2.0 and GAM. Please note that KAN + edge scoring functions other than L1 are **not** baselines; they have been included for edge-scoring comparisons only.

Table 3: Comparison of explanation metrics across datasets (mean ± SD).

| Method | InsAUC ↑ | DelAUC ↓ | Suff ↓ |
|---|---|---|---|
| **OASIS (Acc: 93.24%, AUC: 95.74)** | | | |
| KAN + max | 0.823 ± 0.004 | 0.703 ± 0.035 | 0.099 ± 0.080 |
| Path-Attr. + max | 0.837 ± 0.015 | 0.331 ± 0.080 | 0.103 ± 0.076 |
| GAM | 0.860 ± 0.048 | 0.366 ± 0.161 | 0.046 ± 0.071 |
| KAN 2.0 | 0.845 ± 0.030 | 0.354 ± 0.101 | 0.087 ± 0.073 |
| Path-Attr. + l1 | 0.853 ± 0.029 | 0.274 ± 0.049 | 0.041 ± 0.057 |
| Path-Attr. + l2 | 0.865 ± 0.049 | 0.282 ± 0.020 | 0.068 ± 0.064 |
| Path-Attr. + std | 0.769 ± 0.033 | 0.218 ± 0.004 | 0.042 ± 0.068 |
| Path-Attr. + l1_max | 0.835 ± 0.036 | 0.271 ± 0.016 | 0.082 ± 0.054 |
| Path-Attr. + recursive | 0.847 ± 0.019 | 0.299 ± 0.070 | 0.087 ± 0.080 |
| KAN + l1 | 0.839 ± 0.042 | 0.287 ± 0.024 | 0.072 ± 0.058 |
| KAN + l2 | 0.846 ± 0.017 | 0.454 ± 0.075 | 0.066 ± 0.092 |
| KAN + std | 0.610 ± 0.070 | 0.213 ± 0.006 | 0.348 ± 0.152 |
| KAN + l1_max | 0.692 ± 0.053 | 0.188 ± 0.013 | 0.614 ± 0.031 |
| KAN + recursive | 0.831 ± 0.008 | 0.466 ± 0.070 | 0.068 ± 0.084 |
| **ADNI (Acc: 81.85%, AUC: 91.30)** | | | |
| Path-Attr. + l1_max | 0.678 ± 0.025 | 0.347 ± 0.098 | 0.221 ± 0.029 |
| KAN 2.0 | 0.716 ± 0.033 | 0.439 ± 0.034 | 0.326 ± 0.124 |
| KAN + l2 | 0.746 ± 0.060 | 0.364 ± 0.041 | 0.175 ± 0.172 |
| GAM | 0.746 ± 0.026 | 0.506 ± 0.048 | 0.269 ± 0.136 |
| Path-Attr. + l1 | 0.698 ± 0.028 | **0.310 ± 0.081** | 0.222 ± 0.030 |
| Path-Attr. + l2 | 0.758 ± 0.044 | 0.380 ± 0.051 | 0.209 ± 0.143 |
| Path-Attr. + std | 0.631 ± 0.021 | 0.385 ± 0.069 | 0.211 ± 0.029 |
| Path-Attr. + max | 0.749 ± 0.059 | 0.364 ± 0.041 | 0.201 ± 0.134 |
| Path-Attr. + recursive | 0.740 ± 0.064 | 0.356 ± 0.039 | 0.217 ± 0.168 |
| KAN + l1 | 0.666 ± 0.016 | 0.376 ± 0.075 | 0.221 ± 0.029 |
| KAN + std | 0.631 ± 0.021 | 0.392 ± 0.058 | 0.267 ± 0.036 |
| KAN + max | 0.745 ± 0.061 | 0.354 ± 0.040 | 0.196 ± 0.165 |
| KAN + l1_max | 0.658 ± 0.014 | 0.399 ± 0.077 | 0.222 ± 0.030 |
| KAN + recursive | 0.746 ± 0.072 | 0.336 ± 0.042 | 0.229 ± 0.174 |
| **Mendeley (Acc: 91.25%, AUC: 96.62)** | | | |
| KAN 2.0 | 0.781 ± 0.019 | 0.554 ± 0.027 | 0.301 ± 0.034 |
| Path-Attr. + max | 0.773 ± 0.012 | 0.513 ± 0.067 | 0.265 ± 0.093 |
| Path-Attr. + std | 0.679 ± 0.065 | 0.332 ± 0.020 | 0.261 ± 0.073 |
| GAM | 0.767 ± 0.037 | 0.399 ± 0.090 | 0.293 ± 0.113 |
| Path-Attr. + l1 | 0.649 ± 0.080 | **0.296 ± 0.053** | 0.504 ± 0.103 |
| Path-Attr. + l2 | **0.791 ± 0.016** | 0.463 ± 0.099 | **0.226 ± 0.033** |
| Path-Attr. + l1_max | 0.671 ± 0.073 | 0.300 ± 0.047 | 0.398 ± 0.087 |
| Path-Attr. + recursive | 0.778 ± 0.026 | 0.431 ± 0.097 | 0.247 ± 0.096 |
| KAN + l1 | 0.710 ± 0.080 | 0.401 ± 0.053 | 0.444 ± 0.102 |
| KAN + l2 | 0.730 ± 0.011 | 0.426 ± 0.063 | 0.484 ± 0.051 |
| KAN + std | 0.652 ± 0.074 | 0.320 ± 0.011 | 0.351 ± 0.132 |
| KAN + max | 0.758 ± 0.016 | 0.444 ± 0.064 | 0.443 ± 0.122 |
| KAN + l1_max | 0.683 ± 0.093 | 0.372 ± 0.048 | 0.501 ± 0.049 |
| KAN + recursive | 0.755 ± 0.032 | 0.402 ± 0.124 | 0.316 ± 0.095 |
| **MNIST (Acc: 99.41%, AUC: 99.89)** | | | |
| Path-Attr. + l2 | **0.964 ± 0.003** | 0.293 ± 0.022 | **0.052 ± 0.015** |
| KAN + std | 0.906 ± 0.023 | 0.168 ± 0.046 | 0.153 ± 0.035 |
| KAN + recursive | 0.936 ± 0.020 | 0.250 ± 0.053 | 0.111 ± 0.044 |
| GAM | 0.956 ± 0.004 | 0.380 ± 0.008 | 0.103 ± 0.013 |
| KAN 2.0 | 0.911 ± 0.010 | 0.326 ± 0.111 | 0.412 ± 0.164 |
| Path-Attr. + l1 | 0.756 ± 0.012 | 0.159 ± 0.113 | 0.490 ± 0.003 |
| Path-Attr. + std | 0.753 ± 0.006 | **0.114 ± 0.007** | 0.109 ± 0.013 |
| Path-Attr. + max | 0.944 ± 0.004 | 0.269 ± 0.009 | 0.161 ± 0.015 |
| Path-Attr. + l1_max | 0.917 ± 0.002 | 0.228 ± 0.007 | 0.114 ± 0.039 |
| Path-Attr. + recursive | 0.938 ± 0.014 | 0.264 ± 0.023 | 0.143 ± 0.027 |
| KAN + l1 | 0.924 ± 0.012 | 0.240 ± 0.012 | 0.111 ± 0.015 |
| KAN + l2 | 0.941 ± 0.005 | 0.268 ± 0.006 | 0.315 ± 0.015 |
| KAN + max | 0.932 ± 0.014 | 0.256 ± 0.024 | 0.128 ± 0.034 |
| KAN + l1_max | 0.930 ± 0.015 | 0.248 ± 0.013 | 0.062 ± 0.016 |

## A.4 Tabulated ROIs identified across 15 runs

Table 4 presents an example set of the top 18 regions overall. Not all presented grey and white matter regions are tabulated, as this is a combined tabulation of both regions, and we see that white matter regions are highlighted with a greater magnitude than grey matter regions. We performed a one-dimensional clustering analysis (k-means, $k = 4$) on the mean values across all ROIs. This procedure consistently separated the ROIs into four natural clusters, with an apparent separation between a low-intensity cluster (centered at approximately 0.41) and a higher-intensity cluster beginning near 0.52.

Regions presented include the corpus callosum, corona radiata, and posterior thalamic radiations, alongside subcortical and temporal regions such as the dorsal caudate and TE1/TE1.2 auditory cortex. These structures exhibited both high mean importance values and variability patterns consistent with differences reported in AD populations. Importantly, the retained pattern aligns with established neuroanatomical vulnerability profiles observed in Alzheimer's-related cohorts, supporting the biological plausibility of the model's feature ranking.

This clustering procedure therefore ensures that the retained ROIs represent the prominent and anatomically interpretable contributors to the model, while excluding regions that fall within the lower-intensity portion of the distribution.

Table 4: Example ROIs sorted in descending order of the Mean value.

| Rank | ROI label | Mean | SD | CV (%) |
|------|-----------|------|------|--------|
| 1 | WM: Body of corpus callosum | 2.307 | 0.751 | 32.53 |
| 2 | WM: Genu of corpus callosum | 1.986 | 0.635 | 31.98 |
| 3 | WM: Splenium of corpus callosum | 1.099 | 0.333 | 30.31 |
| 4 | WM: Anterior corona radiata L | 0.886 | 0.268 | 30.24 |
| 5 | WM: Posterior thalamic radiation (optic radiation) R | 0.836 | 0.169 | 20.20 |
| 6 | WM: Posterior thalamic radiation (optic radiation) L | 0.827 | 0.209 | 25.32 |
| 7 | GM: dCa_R | 0.759 | 0.144 | 18.91 |
| 8 | WM: Superior corona radiata L | 0.758 | 0.188 | 24.86 |
| 9 | WM: Anterior corona radiata R | 0.737 | 0.154 | 20.91 |
| 10 | GM: dCa_L | 0.687 | 0.109 | 15.87 |
| 11 | GM: TE1.0/TE1.2_L | 0.672 | 0.063 | 9.42 |
| 12 | WM: Superior corona radiata R | 0.640 | 0.109 | 16.97 |
| 13 | WM: Middle cerebellar peduncle | 0.630 | 0.136 | 21.61 |
| 14 | GM: A10m_R | 0.571 | 0.016 | 2.76 |
| 15 | GM: A10m_L | 0.568 | 0.019 | 3.39 |
| 16 | WM: Posterior corona radiata L | 0.562 | 0.082 | 14.67 |
| 17 | WM: Posterior corona radiata R | 0.543 | 0.095 | 17.45 |
| 18 | GM: TE1.0/TE1.2_R | 0.525 | 0.030 | 5.71 |

