# OpenReview forum: "Interpreting Kolmogorov-Arnold Networks in Neuroimaging: A Path-Based Attribution Framework"
_TMLR — Accepted by TMLR_

### Review · Reviewer_Rq9g · 2025-12-21

**Summary Of Contributions:**

The authors introduce a path-based attribution framework for Kolmogorov-Arnold Networks (KANs) applied to Alzheimer's disease (AD) detection from neuroimaging data as a global attribution method that traces information flow through all computational paths in a trained KAN to generate population-level importance maps.

**Strengths**
1. Provides global attributions as an intrinsic property of the trained network itself, rather than instance-specific explanations
2. Scalable to high-dimensional neuroimaging data
3. Empirically validated on three public AD datasets (OASIS, ADNI, Mendeley), while identifying clinically meaningful regions-of-interest (ROI).

**Weaknesses**
1. Limited to KAN, which AFAIK is not a widely-adopted classification model for AD
2. Scales well, but not *that* well, as Section 4.1 only analyzes the middle sagittal slice from the ADNI.
    + It can be argued that the costly part is training a feed-forward KAN, instead of running the global attribution method. However, this is at least partially because the method isn't well-defined for convolutional/volumetric or other KAN variants, which could be necessary when working with high-dimensional data
3. Only works for binary classification, but unable to produce class-wise attribution map
4. Partially supported claims (see below)

**Audience:**

No

**Audience Explanation:**

The paper is strongly coupled to both KAN (since it relies on the fact that each node only does addition) and neuroimaging data (since the method has many moving parts, e.g., L2 scoring, tanh normalization, absolute edge score magnitudes, and they all had to be handpicked by the authors based on their performance on neuroimaging data). I don't think many individuals in TMLR's audience would be interested in the narrow intersection field between KAN and neuroimaging.

**Claims And Evidence:**

No

**Claims Explanation:**

1. The path-based method only marginally outperforms GAM + IG, KAN 1, and KAN 2 in terms of Insertion/Deletion AUC and Sufficiency (Table 1), and the error bar of many methods are overlapping with each other, so the signal isn't very clear.

1. Insertion/Deletion AUC are problematic by themselves, e.g., they forces the model to extrapolate to artificial/unrealistic inputs that are completely out-of-distribution, and they cannot capture the interaction between features because they add/remove each on individually. One alternative is to run generate simulated datasets with known/synthetic signal and noisy features, and see whether the path-based method can identify the signal features correctly.

1. Many design choices are only backed up by informal/heuristic arguments and experiment result. For example, the authors are using L2 instead of L1 edge scoring function because "[...] this L2 measure avoids underestimating important edges that display a lower variance, penalizes sporadic large activations, and provides a more faithful signal for path-level attribution," but it's not uncommon for L1 to outperform L2 in Table 3.
    + The same goes for tanh normalization, and absolute vs signed edge score magnitudes, which don't even have ablation studies.

1. While KAN achieves test accuracies of 93.24%, 81.85% and 91.25% on OASIS-1, ADNI and Mendeley, we can easily find models that greatly outperforms KAN on test datasets, e.g, a CNN with >99% test accuracy on ADNI: https://www.nature.com/articles/s41598-024-53733-6 . As the authors pointed out themselves, "[t]he attribution maps would be accurate only if the model is able to accurately perform the task at hand."

**Requested Changes:**

1. Expand the scope of the path-based attribution method to domains other than neuroimaging.

1. Clarify the advantage of global attributions over local attributions. In particular, it is known that global attribution is *different* than averaged local attribution, but how is it *better* than the latter? Moreover, how can global attribution reveal biases or spurious correlation in a way that (averaged) local attribution can't?

1. Make Table 3 clearer. The text says "Best performance per dataset and metric appears in bold", but some cases the best performance isn't in bold. For example, on the `OASIS` dataset, the best performance is `KAN + l1_max` with `DelAUC = 0.188 ± 0.013` but the bold item is `Path-Attr. + l1` with `DelAUC = 0.274 ± 0.049`, a significantly higher/worse value. Moreover, the `Suff` and `Comp` columns don't have anything in bold, while the `Prec@5` column has two methods both in bold.

---

> ### Author Response · Authors · 2026-01-30
> **Response to Reviewer Rq9g**
>
> Thank you so much for the detailed feedback and thoughtful engagement with our work! We address the requested changes first, followed by responses to other concerns.
>
> ---
>
> ### **Requested Changes**
>
> #### **1. Expand the scope of the path-based attribution method to domains other than neuroimaging**
>
> Thank you for this request, we have clarified and expanded upon the intended scope and audience of the paper towards the end of the Introduction. We have also explicitly clarified that the choice of edge-scoring is dataset dependent (Section 3), and that the proposed path-based attribution framework can be applied to any problem where a KAN can perform reasonably well with any number of classes. We choose to illustrate the performance of the framework on neuroimaging data due to its challenging high-dimensional nature and relevance for global attribution with its spatial correlations and population-level variability.
>
> We include MNIST as a proof-of-concept sanity check and an example beyond neuroimaging with several classes. We have revised Section 1 to explicitly position the contribution as a methodological framework for global attribution in models with explicit information flow, evaluated in neuroimaging because it is a domain where such methods are both technically demanding and clinically useful. This, along with KANs being a well-cited work, we hope this focus remains well aligned with TMLR's audience, particularly readers interested in interpretability and attribution beyond instance-level explanations.
>
> #### **2. Clarify the advantage of global attributions over local attributions**
>
> Thank you for highlighting this, we're glad to clarify and have expanded upon the same in the Introduction and Related Literature, highlighted in blue. We expanded upon the related literature in Section 2 to clarify that, as discussed and proven in prior work (e.g., Global Attribution Mapping [1], SAGE [2]), averaged local attributions conflate intrinsic feature importance with the empirical data distribution. Features that vary widely across samples can receive high average local scores even if the model does not rely on them causally.
>
> We also clarified in the Introduction that global attribution maps aid model debugging by revealing biases or spurious correlations by highlighting the point of focus of the model, allowing users to identify, with known ground-truth markers, if the model is identifying appropriate features. The proposed path-based attribution is computed directly from the trained network parameters and activations, independent of post-hoc aggregation over individual inputs. This allows it to show systemic effects in global maps, even when such patterns are not dominant in individual predictions. For example, if a model consistently routes information through a particular subnetwork or feature interaction across all inputs, this dependency is revealed by global path attribution even if its effect is subtle at the instance level. Averaged local attributions may wash out such effects due to sample-level variability.
>
> Additionally, computing averaged local attributions or implementing GAM requires access to training data post-hoc, with compute time scaling quadratically with the number of training samples. Our method needs only saved activation information and scales linearly with model size.
>
> We have revised Section 2 to make this distinction clearer.
>
> [1] Covert, I., Lundberg, S. M., & Lee, S. I. (2020). Understanding global feature contributions with additive importance measures. Advances in neural information processing systems, 33, 17212-17223.
>
> [2] Ibrahim, M., Louie, M., Modarres, C., & Paisley, J. (2019, January). Global explanations of neural networks: Mapping the landscape of predictions. In Proceedings of the 2019 AAAI/ACM Conference on AI, Ethics, and Society (pp. 279-287).
>
> #### **3. Make Supplementary Table 3 clearer**
>
> Thank you for pointing this out! We've corrected all formatting inconsistencies and now consistently highlight the best-performing values per dataset and metric, and removed bolding if there is no statistically winning metric. We also simplified Table 3 by removing redundant metrics (such Comprehensiveness, AOPC, Prec@5 and Faith), retaining only Insertion AUC, Deletion AUC, Sufficiency, which were discussed in the main paper.

---

> ### Author Response · Authors · 2026-01-30
> **Response to Reviewer Rq9g (Contd.)**
>
> ### **Weakness 1: Limited to KAN, not widely adopted for AD**
>
> We agree KANs are not yet widely adopted for AD. However, it can be seen that several emerging works explore KANs as discussed in related literature, and we have clarified this scope in the revised introduction and limitations sections accordingly. We study global attribution in models with edge functions and additive nodes where information flow is analytically traceable, which is not possible in CNNs or transformers. KANs provide a uniquely suitable setting for this study.
>
> While we choose to illustrate the performance of the framework on neuroimaging data due to its challenging nature as mentioned before, the framework is agnostic to application domain and applies to any task where a feedforward KAN can be properly trained.
>
> ---
>
> ### **Weakness 2: Scalability due to costly KAN training**
>
> We agree training a feed-forward KAN is computationally expensive. Our scalability claim refers to attribution computation after training. Our method's cost is independent of dataset size and scales linearly with network depth, unlike approaches like GAM (quadratic in samples). Once a KAN is trained, attribution can be computed efficiently and reused without retraining or re-accessing data, as we have done on the OASIS dataset with each input consisting ~900k voxels. We have clarified this in the revised discussion.
>
> For convolutional or hybrid KAN variants, the feasibility depends on the specific form of additive structure; complementary attribution methods may be required. Thank you for pointing this out, and we view this integration as important future work.
>
> ---
>
> ### **Weakness 3: Only binary classification; no class-wise attribution**
>
> We have clarified that the framework is not limited to binary classification, and we demonstrate a multi-class application on MNIST (Table 1, 99.41% accuracy). We clarify on how this extends to multi-class applications in Section 3.2.
>
> However, in this study, we focus primarily on global attributions since the goal is to understand subtle population-level biomarkers and represent the model’s learnt information as a comprehensive map onto input space.  We also acknowledge, in Limitations, that faithful class-wise global attribution is challenging in fully connected KANs. Because edge functions are learned jointly across the full training distribution, restricting path aggregation to a single output neuron does not guarantee disentangled explanations, and we have explicitly discussed this in the Limitations section.

---

> ### Author Response · Authors · 2026-01-30
> **Response to Reviewer Rq9g (Contd..)**
>
> ### **Claim Support Issue 1: Marginal improvement over baselines**
>
> Thank you for this observation. We performed paired Wilcoxon signed-rank tests across 15 independent runs and found that the proposed framework outperforms the baselines on 31 out of 36 benchmarks (i.e., all the combinations of baselines + datasets + metrics) at a CI of 95%. We have added significance indication to Table 1 and rewritten claims to state the approach performs comparably to or better than baselines, provided an appropriate edge-scoring metric is identified.
>
> ---
>
> ### **Claim Support Issue 2: Insertion/Deletion AUC limitations; need synthetic validation**
>
> Thank you for this suggestion. We added a *synthetic experiment* in Appendix A.2:
> We generate 40×40 images with triangles. Class label depends on presence/absence of a hole within a fixed predefined subsection (Ground Truth(GT)), the only causally relevant region, and noisy region is the triangle which is stretched and squeezed in 70% of the samples. Other pixels are visually salient but non-informative (structured noise).
>
> *Expectation:* Because class difference depends exclusively on the subsection containing the hole, a faithful attribution method is expected to assign high importance to the entire subsection and near-zero importance to the rest of the image, including other parts of the triangle that are visually prominent but non-informative. We expect the framework to identify the predefined subsection whilst ignoring the background triangle.
>
> *Result:* Path-based attribution precisely highlights only the GT subsection while ignoring noisy regions. This demonstrates the method correctly recovers causal features, and we can verify the GT as the data-generating process is known.  As such, this experiment provides complementary qualitative evidence that the proposed method can identify true signal features, whilst Insertion/Deletion AUC are metrics we use to quantify the quality of attribution maps.
>
> This complements MNIST visualizations in A.2 and ROI analysis (Sections 5.4–5.5) for attribution validation identifying GT regions.
>
> ---
>
> ### **Claim Support Issue 3: Design choices lack ablation studies**
>
> Thank you for this observation. We have reframed our contributions to specify that the path-based attribution framework can be applied with any meaningful edge-scoring function. Our experiments show the choice is dataset-dependent, as highlighted by Reviewer 74hz. We have now modified Section 3.1 to categorically state that no single edge-scoring function is universally optimal and the edge-scoring function is data-dependent.
>
> **Tanh ablation:** Removing tanh yields negligible differences (ΔInsAUC = 0.002, ΔDelAUC = 0.008, ΔSuff = -0.037 averaged across edge-scoring metrics). We adopted tanh following the original KAN implementation [1] to prevent edge-score explosion, and have clarified our motivation regarding the same.
>
> [1] https://github.com/KindXiaoming/pykan
>
> ---
>
> ### **Claim Support Issue 4: Lower accuracy than CNN baselines**
>
> Thank you for raising this important point. Our objective was to study population-level attribution as an intrinsic property of trained networks, and to also not rely on data post-hoc. We focus on KANs because their edge functions and addition-multiplication on nodes structure enable attribution mechanisms that CNNs cannot provide. Considering the sensitive nature of neuroimaging and ROI-analysis, we employ minimal data augmentation and do not use augmentation such as synthetic data addition (e.g., [1]). Additionally, CNNs also do not support network-intrinsic global attribution similar to the proposed path-based framework.
>
> Our contribution addresses a different axis of the accuracy-interpretability trade-off. Our ROI validation (Sections 5.4–5.5) confirms the model learned clinically meaningful patterns consistent with established AD literature, demonstrating biological plausibility. We have also clarified the accuracy caveat in the Limitations section.
>
> [1] El-Assy, A.M., Amer, H.M., Ibrahim, H.M. et al. A novel CNN architecture for accurate early detection and classification of Alzheimer’s disease using MRI data. Sci Rep 14, 3463 (2024). https://www.nature.com/articles/s41598-024-53733-6
>
> ---
>
> Thank you once again for the detailed and thorough feedback! We sincerely believe this input has helped us significantly better our work, and we hope our response was satisfactory. We would be happy to address any further concerns as well.

---

### Review · Reviewer_74hz · 2026-01-15

**Summary Of Contributions:**

This paper proposes a path-based attribution framework specifically designed for Kolmogorov-Arnold Networks (KANs) to address the limitations of local, instance-specific interpretability in neuroimaging. The primary contributions include:
1. Propose a global attribution method that scores network edges using the L2 norm of learned spline and base functions.
2. Propose an efficient implementation using matrix propagation that reduces computational complexity from exponential to linear relative to network depth, making voxel-level global attribution feasible for high-dimensional data.
3. Evaluate the proposed framework and compared with baselines methods on three major Alzheimer’s Disease datasets (OASIS, ADNI, Mendeley) comprising.

**Audience:**

Yes

**Audience Explanation:**

The paper would be of interest for researchers interested in KAN, explainable AI and AD neuroimaging.

**Broader Impact Concerns:**

1. The authors claim that the generated attribution maps "support clinical validation" and help gain "clinical trust". However, there is a significant risk that clinicians may over-rely on these "interpretable" maps, treating them as ground-truth biomarkers rather than decision-support aids. The authors should explicitly state that these maps are intended for research and decision-support, not as standalone diagnostic evidence.
2. The study relies on well-known public datasets. These cohorts are known to have demographic biases, often underrepresenting certain racial, or socioeconomic groups. Because the proposed method produces a global map based on the training population, this could lead to a diagnostic framework that is less accurate for patients from underrepresented demographics. A broader impact statement should acknowledge these limitations and the need for validation on more diverse, real-world cohorts.

**Claims And Evidence:**

Yes

**Claims Explanation:**

The technical implementation of path-based attribution framework and its linear scalability is clearly demonstrated and mathematically sound. However, the empirical "convincingness" is hampered by
1. Norm Selection: While the $L_2$ norm performs best on most datasets, the $max$ norm is superior on the ADNI dataset for both DelAUC and Suff. The authors claim $L_2$ is more stable, but the performance gap is not addressed in detail. The author fails to clarify whether norm selection is a dataset-dependent hyperparameter or if there is a theoretical justification for prioritizing $L_2$.
2. The performance advantage of the "Path-Attr" method over existing baselines such as GAM is frequently marginal. For example, Path-Attr + $L_2$ achieved an Insertion AUC (InsAUC) of 0.865, representing a negligible improvement over the second-best approach (GAM + IG) at 0.860. In the absence of rigorous statistical significance testing (e.g., p-values or confidence intervals), it is difficult to verify if these gains represent a robust methodological advancement or are simply the result of experimental noise.

**Requested Changes:**

1. Add significance testing to the results in Table 1 to ensure reported differences are meaningful.
2. Provide a qualitative or quantitative discussion on why the $max$ norm outperformed $L_2$ on ADNI. Is it due to data-specific biomarkers or preprocessing differences?
3. Introduce and define the abbreviation “GAM” before its first use.

---

> ### Author Response · Authors · 2026-01-30
> **Response to Reviewer 74hz**
>
> Thank you so much for your thoughtful review and constructive feedback! We greatly appreciate your recognition of the technical soundness and linear scalability of our framework, as well as your valuable suggestions for improving empirical rigor and broader impact considerations. We address each of your points below.
>
> ---
>
> ### **Requested Changes**
>
> **1. Add significance testing to the results in Table 1 to ensure reported differences are meaningful**
>
> Thank you for pointing this out! We performed paired Wilcoxon signed-rank tests across 15 independent runs and found that across all the reported metrics and datasets (3 metrics - InsAUC, DelAUC, Suff; 4 datasets - OASIS, ADNI, Mendeley, MNIST; 3 baselines - GAM, KAN 2.0, KAN), the proposed framework outperforms the baselines on 31 out of 36 benchmarks (i.e., all the combinations of baselines + datasets + metrics) at a CI of 95%.. We have appropriately rewritten claims to state that the approach performs comparably to or better than other baselines, provided an appropriate edge-scoring metric is identified to work on the dataset.
>
> ---
>
> **2. Provide a qualitative or quantitative discussion on why the max norm outperformed L2 on ADNI. Is it due to data-specific biomarkers or preprocessing differences?**
>
> Thank you for this observation. We have added a quantitative discussion on the performance between inter- and intra-path-attribution metrics and baselines in Section 5.1. We have also reframed our contributions and now specify that **the choice of edge-scoring is dataset-dependent**, and that the path-based attribution framework can be applied with any meaningful edge-score. Across the paper in Sections 3.1 and 5.1, we clarify that edge-scoring metrics can be replaced by any appropriate edge-scoring function to work in tandem with the proposed path-based attribution. The choice of edge-scoring was purely empirical and we have rewritten our claims to specify that **no single edge-scoring function is universally optimal**. We interpret this difference as dataset-dependent behavior, and we do not claim a theoretical guarantee that L2 should dominate in all settings.
>
> ---
>
> **3. Introduce and define the abbreviation "GAM" before its first use**
>
> Thank you for pointing this out! **GAM (Global Attribution Mapping)** is now defined before its first use.
>
> ---
>
> ### **Broader Impact Concerns**
>
> **Concern 1: Over-reliance on attribution maps as ground-truth biomarkers**
>
> We added a separate section, **Broader Impact Concerns**, which explicitly discusses automation bias and states that generated attribution maps are intended as *research and decision-support tools, not as standalone diagnostic evidence or ground-truth biomarkers*. We now emphasize that these maps are meant to support hypothesis generation, model auditing, and population-level analysis, and that any clinical use must occur in conjunction with established diagnostic protocols and expert judgment.
>
> ---
>
> **Concern 2: Demographic biases in public datasets**
>
> The added section also acknowledges that the public neuroimaging datasets used in this study (OASIS, ADNI, Mendeley) are known to exhibit *demographic and socioeconomic biases*. Because our method produces global attributions derived from the training population on the aforementioned datasets, these maps may underrepresent or mischaracterize patterns present in underrepresented groups. We have added a broader impact discussion noting that *validation on more diverse, real-world cohorts is necessary* before any clinical deployment and that global attribution methods should be interpreted in light of the populations on which they are trained.
>
> ---
>
> Thank you once again for your detailed and constructive feedback! We believe your input has significantly strengthened the rigor and responsibility of our work, and we hope our revisions address your concerns comprehensively. We would be happy to address any further questions as well.

---

### Review · Reviewer_K6ZF · 2026-01-17

**Summary Of Contributions:**

Here are some key contributions in top of KAN's

* They built a path-based framework to trace exactly how the AI makes decisions
* Energy-Based Scoring: Used L_2 energy norms to figure out which parts of the AI's internal functions are doing the most work
* Created a matrix trick so this works on massive 3D brain scans in seconds instead of years
* Produced super-detailed maps showing exactly which tiny 3D pixels (voxels) the AI is looking at.
* Showed the AI is actually smart by proving it focuses on the same brain regions doctors already worry about, like the corpus callosum and caudate nuclei.

**Audience:**

Yes

**Audience Explanation:**

This paper moves the needle on both Kolmogorov-Arnold Network theory and practical AI explainability. It solves a massive bottleneck in neuroimaging by introducing a matrix-based method that handles the huge dimensionality of 3D brain scans, which usually crashes standard path-counting algorithms. Beyond just math, it proves that these newer, flexible networks can actually be trusted in a clinical setting by showing their internal logic aligns perfectly with established human brain anatomy. By turning a black box into a transparent map of medical biomarkers, it provides a blueprint for how AI can gain regulatory approval and clinical trust in high-stakes fields like neuroscience

**Broader Impact Concerns:**

The main ethical concern is the risk of "automation bias," where clinicians might treat the AI’s global importance maps as absolute truth rather than a suggestive tool, potentially leading to misdiagnosis if the model relies on spurious correlations. Since the framework currently lacks class-wise distinction, there is also a risk of misinterpreting general brain aging patterns as specific Alzheimer's pathology.

**Claims And Evidence:**

Yes

**Claims Explanation:**

he claims are solid because the model actually hits high marks, like 93.24% accuracy on the OASIS dataset and 99.4% on MNIST. The authors proved their maps are "faithful" using math tests called Insertion and Deletion AUC, where their method consistently beat older ones like GAM and KAN 2.0. This means when the AI says a brain region is important, it’s not just guessing; removing those areas actually tanks the model's confidence.

The evidence is even more convincing because the AI's internal thoughts match real medical science. It consistently zeroed in on the corpus callosum and caudate nuclei across 15 different test runs, which are exactly the spots doctors look at for Alzheimer's. With a massive testing pool of over 7,400 image acquisitions.

**Requested Changes:**

* Explicitly state that the attribution maps are only as reliable as the model's test accuracy, which is currently around 93% for OASIS.
* Showing separate "Healthy" vs. "Diseased" patterns would be very helpful the current framework only produces one combined global map

---

> ### Author Response · Authors · 2026-01-30
> **Response to Reviewer K6ZF**
>
> Thank you so much for your incredibly thoughtful and enthusiastic review! We are delighted that you found the technical contributions solid and recognized the clinical relevance of our work. Your feedback on reliability and class-wise attribution has helped us clarify important limitations and ethical considerations. We address each of your points below.
>
> ---
>
> ### **Requested Changes**
>
> **1. Explicitly state that the attribution maps are only as reliable as the model's test accuracy, which is currently around 93% for OASIS**
>
> We thank the reviewer for this suggestion and fully agree. As noted in the manuscript, and under Limitations, attribution maps are only meaningful to the extent that the underlying model has learned a reliable decision function. We state that the generated attribution maps are contingent on model performance (e.g., ~93% test accuracy on OASIS), and that incorrect or weakly trained models would yield unreliable explanations. To further emphasize this point, we have reinforced this statement in the Limitations section, clearly noting that attribution quality is bounded by classification accuracy and should be interpreted accordingly.
>
> ---
>
> **2. Showing separate "Healthy" vs. "Diseased" patterns would be very helpful—the current framework only produces one combined global map**
>
> In principle, pseudo-class-wise attribution can be obtained by restricting path aggregation to the output neuron corresponding to a specific class. However, in fully connected KANs, the edge functions feeding into each output neuron are learned jointly from the entire training distribution, not from class-specific data alone. As a result, such class-restricted path analysis may not yield faithful or disentangled class-conditional explanations, and we therefore avoid presenting them as reliable evidence. We have added a discussion of this limitation explicitly in Section 5.3, noting the challenges and faithfulness concerns. We view robust class-conditional global attribution in fully connected KANs as an important direction for future work, rather than a resolved component of the current framework.
>
> ---
>
> ### **Broader Impact Concerns**
>
> **The main ethical concern is the risk of "automation bias," where clinicians might treat the AI's global importance maps as absolute truth rather than a suggestive tool, potentially leading to misdiagnosis if the model relies on spurious correlations. Since the framework currently lacks class-wise distinction, there is also a risk of misinterpreting general brain aging patterns as specific Alzheimer's pathology**
>
> We added a separate section, **Broader Impact Concerns**, which explicitly discusses automation bias and states that generated attribution maps are intended as research and decision-support tools, not as standalone diagnostic evidence or ground-truth biomarkers. We now emphasize that these maps are meant to support hypothesis generation, model auditing, and population-level analysis, and that any clinical use must occur in conjunction with established diagnostic protocols and expert judgment. The added section also acknowledges that the public neuroimaging datasets used in this study (OASIS, ADNI, Mendeley) are known to exhibit demographic and socioeconomic biases. Because our method produces global attributions derived from the training population and does not produce class-wise maps, these maps may underrepresent or mischaracterize patterns present in underrepresented groups and aging biomarkers. We have added a broader impact discussion noting that validation on more diverse, real-world cohorts is necessary before any clinical deployment and that global attribution methods should be interpreted in light of the populations on which they are trained.
>
> ---
>
> Thank you once again for your detailed and encouraging feedback! We are grateful for your recognition of the clinical and methodological contributions, and we believe your insights have strengthened both the technical rigor and ethical responsibility of our work. We would be happy to address any further questions as well.

---

### Review · Reviewer_3KX1 · 2026-01-20

**Summary Of Contributions:**

The paper focuses on the problem of interpretability/explainability of neural network models in a medical context. Specifically, the paper builds upon the Kolmogorov-Arnold Networks (KAN) and proposes a path-based attribution framework.

The paper demonstrates that ...

- the proposed method can be scaled to voxel-level rather than ROI-level
- the proposed method performs well on several metrics for three Alzeheimer's Disease (AD) datasets.

The paper is generally well-written, well-structured, and easy to follow. The method and math are sound. However, there are no theoretical justification or analysis of the proposed method.

**Additional Comments:**

A few minor comments:

- The paper states "With KANs being proposed as interpretable networks, there have been applications in medicine. Dong et al. (2025) used a KAN-based Graph Convolutional Networks (GCNs) for Alzheimer’s disease diagnosis on the ADNI dataset, with their GCN-KAN achieved 62.6% over standard GCNs at 57.4%."
	- What are the percentages?  Accuracy or some other metrics?

- "We use a simple feedforward deep KAN, each with about 6-10 hidden layers and run the attributions on the fully trained model."
	- Please be more precise :-)

- Base activation function seems to change notation from b(x) to sigma(x). Is there any significance to this?

- "We score the paths as the absolute of the tanh of the product of edge-scores on the path.". But there is no absolute value in eq. (2). Am I missing something?

**Audience:**

Yes

**Audience Explanation:**

The problem of interpreting and explaining decisions from black box models is getting increasingly important for both researchers and practitioners in ML and the medical domain.

**Claims And Evidence:**

Yes

**Claims Explanation:**

There is no theoretical justification or analysis for the proposed method, but the authors provide empirical evidence showing that the proposed method outperforms competing baseline with a small margin.

The paper emphasises the significance of the deployed L2-norm as a contribution. However, the significance of this is not entirely clear to me.

**Requested Changes:**

Overall, the paper is well-written, well-structured and easy to follow. However, I believe there a several areas, where the paper could benefit from a clarification and/or additional discussion.

1) It would be great if the author could add a single paragraph about the "overall workflow of using the method". The way I understand the method is as follows. Suppose you want to build a model for AD's given some dataset, then you identify an appropriate architecture for the problem/dataset. When you are happy with your architecture, you reparametrize all edges of the network using the base + split parametrization and retrain the network. Finally, you compute the attribution maps of interest. Is this correct?

2) The main performance metrics for the experiments are the "Insertion AUC", "Deletion AUC", and "Sufficiency", whose informal definition is repeated twice in section 4.1 and 4.2 It would be great if the author could replace one of the informal descriptions with a proper mathematical definition for clarity.

3) The proposed method is spline-based, but there is are results or discussion for the sensitivity the results are to the specific spline-configuration (cubic B-splines with 5 knots per dimension). Is 5 knots always sufficient? Moreover, section A.1 mention a "noise scale" of 0.1, but it is unclear to me what this refers to.     Finally, Appendix A.1 states that the spline grid is defined to be [-1, 1] and the input is scaled accordingly. But it is unclear to me what happens on the subsequent layers. Please clarify these details.

4) The paper makes heavy use of the word "faithful", but does not provide a clear definition (or a clear reference).

5) The paper is centered around the application to medical applications (AD) and states that local attributions methods can be useful for "individual cases, such as tumor detection", but less so for studying AD. Could you make elaborate a bit on this for the average non-medical ML-audience like me?

6) It is unclear how the method is deployed for other problems than binary classification. How does the method apply to multi-class classification like the MNIST problem?

7) It would also be great with a more in-depth discussion of the drawback of the proposed method besides the "accuracy-constraint".

---

> ### Author Response · Authors · 2026-01-30
> **Response to Reviewer 3KX1**
>
> Thank you so much for your very thorough and detailed review! We are grateful for your recognition that the paper is well-written, well-structured, and technically sound. Your constructive feedback has helped us clarify several important methodological and notational details. We address each of your points below.
>
> ---
>
> ### **Requested Changes**
>
> **1. Add a single paragraph about the "overall workflow of using the method"**
>
> Thank you for this suggestion. We have added a step-by-step guide in Appendix A.1 describing the workflow, and prerequisites to apply our framework. Given a fully trained KAN with saved activation information on the functions on edges, the proposed path-attribution framework can be applied post-hoc to compute attribution maps without requiring access to the training data. This allows the model to be analyzed independently of the original dataset, facilitating reproducibility and portability. Below is a short descriptive algorithm of the workflow:
>
> > *Input:* Trained feedforward network with i) nonlinear functions are associated with edges, (ii) nodes aggregate incoming signals additively or multiplicatively, (iii) directed paths exist from input features to output neurons, given saved edge functions $\{\phi_{j,k}^{(\ell)}\}$ and model parameters:
> >
> > *Parameters:* $L$ (number of layers), $d_0$ (input dimensionality), $d_\ell$ (width of layer $\ell$), $n = d_L$ (output neurons)
> >
> > *Output:* Global input-level attribution scores $\mathbf{s} \in \mathbb{R}^{d_0}$
> >
> > ---
> >
> > *for* each layer $\ell = 1, \ldots, L$ *do*
> >
> > &nbsp;&nbsp;&nbsp;&nbsp; Construct edge-importance matrix $\boldsymbol{\alpha}^{(\ell)} \in \mathbb{R}^{d_\ell \times d_{\ell-1}}$ with entries:
> >
> > &nbsp;&nbsp;&nbsp;&nbsp; $\alpha_{j,k}^{(\ell)} \leftarrow \tanh(E_{j,k}^{(\ell)})$, &nbsp;&nbsp; where $E_{j,k}^{(\ell)} = g(\phi_{j,k}^{(\ell)})$
> >
> > &nbsp;&nbsp;&nbsp;&nbsp; (where $g(\cdot)$ is edge-scoring function, e.g., L2 norm)
> >
> > *end for*
> >
> > Compute propagated importance matrix:
> >
> > $\mathbf{M} \leftarrow \boldsymbol{\alpha}^{(L)} \cdot \boldsymbol{\alpha}^{(L-1)} \cdots \boldsymbol{\alpha}^{(1)} \in \mathbb{R}^{n \times d_0}$
> >
> > *for* each input feature $i \in \{1, \ldots, d_0\}$ *do*
> >
> > &nbsp;&nbsp;&nbsp;&nbsp; $s_i \leftarrow \frac{1}{n} \sum_{o=1}^{n} |\mathbf{M}_{o,i}|$
> >
> > *end for*
> >
> > Optionally normalize $\mathbf{s}$ for visualization or comparison
> >
> > *return* $\mathbf{s}$
>
> ---
>
> **2. Replace one of the informal descriptions of "Insertion AUC", "Deletion AUC", and "Sufficiency" with a proper mathematical definition**
>
> We have revised the manuscript to replace one of the informal descriptions, i.e., the description under Section 4.2 with explicit mathematical definitions of Insertion AUC, Deletion AUC, and Sufficiency, following standard formulations in the attribution literature for clarity.
>
> ---
>
> **3. Discuss sensitivity to spline configuration (cubic B-splines with 5 knots), clarify "noise scale" of 0.1, and explain spline grid behavior in subsequent layers**
>
> Thank you for pointing out that these implementation details were insufficiently explained. The spline configuration (cubic B-splines with 5 knots per dimension) was configured based on guidance from the original KAN implementation [1] and kept fixed across all experiments to avoid introducing additional hyperparameter variation. We do not claim that 5 knots are universally optimal; rather, they provided stable training and sufficient expressivity in our experiments.
>
> The "noise scale" refers to small additive noise used during spline initialization, as introduced in the original KAN implementation, to encourage smoother function learning and avoid degenerate solutions.
>
> Regarding the spline grid range, subsequent layers operate on activations produced by preceding KAN layers, whose outputs naturally lie within the effective spline support due to the learned base and spline functions. We have clarified these points in Appendix A.1.
>
> [1] https://github.com/KindXiaoming/pykan
>
> ---
>
> **4. Provide a clear definition (or reference) for the word "faithful"**
>
> We agree that the term "faithful" should be defined explicitly. In the revised manuscript, we define faithfulness to denote how closely the produced attribution maps reflect the features the model has learned and utilizes to arrive at its decisions, with attribution quality quantified using perturbation-based metrics such as Insertion AUC, Deletion AUC, and Sufficiency. This definition now appears in the Introduction.

---

> ### Author Response · Authors · 2026-01-30
> **Response to Reviewer 3KX1 (Contd.)**
>
> **5. Elaborate on why local attribution methods are useful for tumor detection but less so for studying AD, for the average non-medical ML audience**
>
> Thank you for suggesting this clarification! We have expanded the discussion to explain that tumors are patient-specific; one patient having a tumor at a particular region does not imply another patient will have a tumor in the same location, making local attributions useful for individual diagnosis. In contrast, neurodegenerative diseases such as Alzheimer's involve diffuse, population-level structural changes that are better captured by global attribution methods. This distinction makes path-attribution particularly useful for group-level studies (e.g., Alzheimer's) versus subject-specific analysis (e.g., brain tumors).
>
> ---
>
> **6. Clarify how the method applies to multi-class classification, such as the MNIST problem**
>
> We have clarified that the proposed framework extends to multi-class classification by computing path-based attributions with respect to each output neuron and aggregating across all output neurons as specified in Equation (2). This produces a single global attribution map that reflects feature importance together across all classes. We applied this approach on MNIST to generate one map across 10 output neurons.
>
> ---
>
> **7. Provide a more in-depth discussion of the drawbacks of the proposed method besides the "accuracy-constraint"**
>
> We agree and have expanded the Limitations section to include additional drawbacks, such as: the lack of inherently faithful class-conditional global attribution in fully connected KANs, sensitivity to architectural choices and the performance of the trained model, and that the proposed path-attribution is better suited to deeper networks due to its path-tracing formulation.
>
> ---
>
> **8. Clarify the significance of the deployed L2-norm as a contribution**
>
> Thank you for pointing this out. We have revised the manuscript to clarify that the L2 norm is not a core theoretical contribution, but rather an empirical instantiation of a local edge-scoring function and that the L2 norm may not always be the best choice. The central contribution is the path-based attribution framework and its efficient matrix-based propagation, which can be applied with any suitable edge-scoring function that captures the contribution of the function at its respective edge. We have softened language that could be interpreted as over-emphasizing L2 and now clearly present it as one effective empirical choice rather than a principled optimum.
>
> ---
>
> ### **Additional Comments**
>
> **1. What are the percentages referring to in "GCN-KAN achieved 62.6% over standard GCNs at 57.4%"?**
>
> Thank you for pointing this out! We have clarified that these percentages refer to classification accuracy.
>
> ---
>
> **2. "We use a simple feedforward deep KAN, each with about 6-10 hidden layers..." — Please be more precise**
>
> We now specify the exact number of hidden layers used for each dataset in the main text under Section 4.2, and have clarified Appendix A.1 for further architectural details.
>
> ---
>
> **3. Base activation function notation changes from b(x) to σ(x). Is there any significance to this?**
>
> Thank you for catching this inconsistency. We have unified the notation for the base activation function, using b(x) consistently throughout the manuscript.
>
> ---
>
> **4. "We score the paths as the absolute of the tanh of the product of edge-scores on the path." But there is no absolute value in Eq. (2). Am I missing something?**
>
> Thank you for pointing this out! We have corrected Equation (2) to include the absolute value, ensuring consistency with the textual description.
>
> ---
>
> Thank you once again for your detailed and constructive feedback! We believe your input has significantly improved the clarity and rigor of our work, and we hope our revisions address your concerns comprehensively. We would be happy to address any further questions as well.

---

### Decision · Action_Editor_MS62 · 2026-03-04

**Recommendation:** Accept as is

**Audience:**

Yes

**Audience Explanation:**

XAI methods is within the ambit of TMLR.

**Claims And Evidence:**

Yes

**Claims Explanation:**

The reviewers have reached consensus that this submission merits publication.